# Decompose a Task into Generalizable Subtasks in Multi-Agent Reinforcement Learning

**Zikang Tian**[1,2,3], **Ruizhi Chen**[4], **Xing Hu**[1,5], **Ling Li**[2,4], **Rui Zhang**[1], **Fan Wu**[2,3,4],
**Shaohui Peng**[4], **Jiaming Guo**[1], **Zidong Du**[1,5], **Qi Guo**[1], **Yunji Chen**[1,2*]

[1]SKL of Processors, Institute of Computing Technology, CAS, Beijing, China
[2]University of Chinese Academy of Sciences, Beijing, China
[3]Cambricon Technologies, Beijing, China
[4]Intelligent Software Research Center, Institute of Software, CAS, Beijing, China
[5]Shanghai Innovation Center for Processor Technologies, SHIC, Shanghai, China
{tianzikang21s, cyj}@ict.ac.cn,

## Abstract

In recent years, Multi-Agent Reinforcement Learning (MARL) techniques have made significant strides in achieving high asymptotic performance in single task. However, there has been limited exploration of model transferability across tasks. Training a model from scratch for each task can be time-consuming and expensive, especially for large-scale Multi-Agent Systems. Therefore, it is crucial to develop methods for generalizing the model across tasks. Considering that there exist task-independent subtasks across MARL tasks, a model that can decompose such subtasks from the source task could generalize to target tasks. However, ensuring true task-independence of subtasks poses a challenge. In this paper, we propose to **d**ecompose a **t**ask in**to** a series of **g**eneralizable **s**ubtasks (DT2GS), a novel framework that addresses this challenge by utilizing a scalable subtask encoder and an adaptive subtask semantic module. We show that these components endow subtasks with two properties critical for task-independence: avoiding overfitting to the source task and maintaining consistent yet scalable semantics across tasks. Empirical results demonstrate that DT2GS possesses sound zero-shot generalization capability across tasks, exhibits sufficient transferability, and outperforms existing methods in both multi-task and single-task problems.

## 1 Introduction

In the last few years, many works in MARL field prospers, including value-based algorithms [26, 25, 20, 27, 30, 39, 10, 17], and policy-based algorithms [6, 36, 35, 11, 24, 7]. However, these works mainly focused on the model's asymptotic performance in a single task, neglecting its transferability across tasks. Training an optimal model on a single task requires millions of interactions with the environment [21, 16], particularly when dealing with large-scale Multi-Agent Systems, whereas transferring the model across tasks can reduce training cost by dozens of times [2, 5]. As the number of tasks increases, the reduction in training costs due to model transfer will become even more significant.

Knowledge reuse is a common approach for model generalization across tasks in the MARL field [4]. Recent knowledge reuse methods for online MARL can be roughly classified into two categories: network-design-based methods and task-embedding-based methods. Network-design-based methods [33, 9, 1] implicitly reuse knowledge extracted from the source task to the target task by constructing

---

*Corresponding author.

37th Conference on Neural Information Processing Systems (NeurIPS 2023).

universal model structure across tasks by utilizing Population Invariant Structure such as Transformer [9] or GNN [1]. However, it's unclear whether the knowledge extracted from the source task is suitable for the target task. Task-embedding-based methods [19, 23, 14] reuse knowledge by calculating task similarity using learned task embeddings that capture task dynamics. However, accurately mapping task dynamics to task embeddings requires numerous tasks as samples. In a word, current knowledge reuse methods for online MARL still have limitations, that is, inefficient knowledge reuse and reliance on a large number of tasks samples. Additionally, we provide a section for related work in Appendix A to introduce these methods more specifically.

Compared with the common knowledge reused in these methods above, there exists an alternative knowledge named task-independent subtasks that can relieve these two limitations. These task-independent subtasks, such as "hit and run", "focuse fire", "disperse and line up", etc, are certainly applicable across tasks [41], improving the efficiency of knowledge reuse. Besides, these task-independent subtasks can be decomposed from few tasks, so as to remove the reliance on a large number of tasks samples. However, ensuring the task-independence of decomposed subtasks is a challenge. Task-independence of subtasks refers to their effectiveness across tasks, which requires two essential properties: **(1) avoiding overfitting to the source task**, **(2) maintaining consistent yet scalable semantics across tasks**. In this paper, we propose DT2GS, a novel framework devoted to generalize model across tasks by decomposing subtasks from the source tasks and endowing them with these two properties necessary for task-independence.

The proposed framework, DT2GS, aims to **d**ecompose a **t**ask in**to** a series of **g**eneralizable **s**ubtasks, as shown in Figure 1. DT2GS is primarily composed of two parts: the scalable subtask encoder and the adaptive action decoder. The scalable subtask encoder assigns a subtask to each agent based on its {subtask, entity-observation} history rather than the {action, observation} history. This approach helps prevent the process of assigning subtasks from overfitting to the source task. The adaptive action decoder leverages the assigned subtasks and current entity-observations to calculate the specific actions for interacting with environment. Within this decoder, the adaptive subtask semantic module ensures that the assigned subtasks have consistent yet scalable semantics across tasks based on their effects on entities. Based on the experimental results, our framework exhibits several desirable properties as follows:

- **zero-shot generalization capability**: the trained model can be effectively depolyed to multiple target tasks without any fine-tuning;
- **robust transferability**: significantly accelerated model convergence on complex target tasks and achieving an average speedup of $100\times$;
- **better asymptotic performance**: achieved state-of-the-art (SOTA) results on multi-task and single-task problems;
- **better subtask interpretability**: decomposed task-independent subtasks with practical behavioral semantics that is consistent yet scalable across tasks.

## 2 Preliminary

### 2.1 Background

In this paper, a fully cooperative multi-agent system (MAS) is described as a decentralized partially observable Markov decision processes (Dec-POMDP) [18], which is defined by a tuple $G = < \mathcal{S}, \mathcal{A}, O, \gamma, n, \mathbb{A}, \Omega, R, P >$. $\mathcal{S}$ is the global state space. $\mathcal{A}$ is the action space shared for all agents. $O$ is the shared individual observation space for all agents. $\gamma$ is the discount factor. $n$ is the number of agents. $\mathbb{A} = \{1, ..., n\}$ is the set of agents our algorithm controlled. At each timestep $t$, agent $i$ obtains an individual observation $o_i^t \in O$ from dynamic environment according to the observation function $\Omega(s^t, i)$. And if we suppose agent $i$ are controlled by policy $\pi$, which takes $o_i^t$ or history individual observations $\tau_i^t$ as input and parameterized by $\theta_i$, agent $i$ will select an action $a_i^t$ according to $\pi(a_i^t|o_i^t)$. Therefore, a joint action $A^t = (a_1, a_2, ..., a_n) \in \mathcal{A}$ will be formed where $a_i$ corresponds to agent $i$. After passing the joint action $A^t$ into the environment, a global reward signal $r_t = R(s^t, A^t)$ shared by all agents will be received and the environment will transmit into next state $s^{t+1}$ according to the transition function $P(s^{t+1}|s^t, A^t) : \mathcal{S} \times \mathcal{A} \times \mathcal{S} \to [0, 1]$. The goal of agents is defined as finding the

policy $\pi$ to maximize the objective function

$$J(\pi) = \mathbb{E}[\sum_{t'=0}^{\infty} \gamma^{t'} R(s^{t'}, A^{t'})] \tag{1}$$

where $\gamma^{t'} R(s^{t'}, A^{t'})$ is the discounted return of all agents.

## 2.2  Generalizable model structure in MARL

For model generalization in MAS, a generalizable model structure is necessary for addressing the problem of varying state/observation/action space ($\mathcal{S}/\mathcal{O}/\mathcal{A}$) across tasks. Here we define the agents controlled by MARL policy and agents built-in tasks as entities. In this paper, we use $n, m, n_{ally}, n_{enemy}$ denotes the number of agents, entities, allies and enemies, respectively. And the following equation holds: $n = n_{ally} + 1, m = n + n_{enemy}$. As demonstrated in ASN [33], an agent's observation $o_i$ can be constructed as a concatenation of $m$ entity-observations: $o_i = [o_{i,1}, o_{i,2}, ..., o_{i,m}]$, where $o_{i,1}$ is the observation of agent $i$ on itself and environment, and the rest are the observations of agent $i$ on other $m - 1$ entities. Additionally, action space $\mathcal{A}$ can be decomposed into two components: $\mathcal{A}^{self}$, which consists of actions affecting the agent itself or the environment, and $\mathcal{A}^{interactive}$, which contains actions that directly interact with other entities. This alignment between entity-observations and actions, which is referred as action semantics, forms the foundation for computing the value or probability of an action based on its aligning entity-observation, leading in a generalizable model structure across tasks.

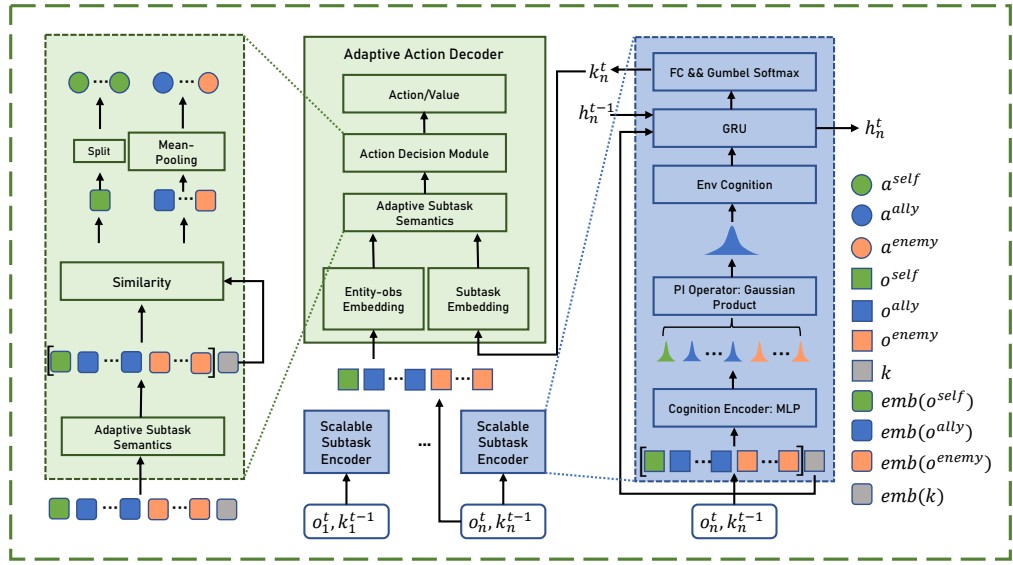

Figure 1: DT2GS Framework. The DT2GS Framework comprises two modules: the Scalable Subtask Encoder and the Adaptive Action Decoder, where the Adaptive Subtask Semantic module serves as the core of the Adaptive Action Decoder. The Scalable Subtask Encoder effectively assigns subtasks to agents without overfitting to the source task. With the Adaptive Subtask Semantic module endowing assigned subtasks with consistent yet scalable semantics across tasks, the action decoder takes the entity-observations and subtasks as inputs to generate actions for interacting with the environment.

## 3  DT2GS Framework

DT2GS primarily consists of two components: the scalable subtask encoder and the adaptive action decoder, as shown in Figure 1. DT2GS begins with the scalable subtask encoder, which assigns a subtask to each agent based on its {subtask, entity-observation} history instead of the {action, observation} history, so as to avoid overfitting to the source tasks. Then the adaptive action decoder calculates the specific actions for interacting with environment by leveraging the assigned subtasks

and current entity-observations. Within this decoder, the adaptive subtask semantic module, which plays a core role, ensures that the assigned subtasks have consistent yet scalable semantics across tasks based on their effects on entities. With the proposed scalable subtask encoder and the adaptive subtask semantic module, DT2GS endows the decomposed subtasks with task-independence, leading model generalizable across tasks. A pseudocode is provided in Appendix H to illustrate the DT2GS framework.

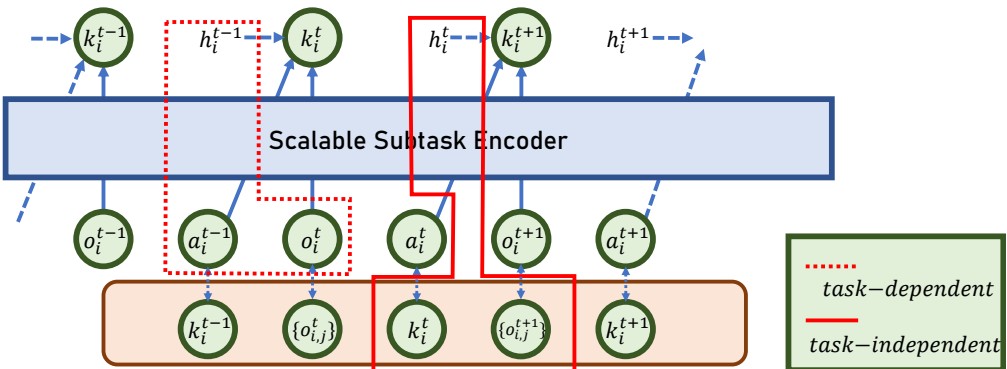

Figure 2: Scalable Subtask Encoder. The Scalable Subtask Encoder assigns subtasks to each agent based on its behavioral history representation, which is depicted by its {subtask, entity-observation} history instead of commonly used {action, observation} history. The red dashed-line box denotes the {action, observation} history, which is task-dependent. Specifically, at each timestep $t$, agent $i$ obtains its subtask $k_i^t$ by utilizing its observation $o_i^t$, last action $a_i^{t-1}$, history latent embedding $h_i^{t-1}$ produced by RNN. On the contrary, the **red solid-line box** denotes the {subtask, entity-observation} history, which is task-independent. Specifically, at each timestep $t + 1$, agent $i$ obtains its subtask $k_i^{t+1}$ by utilizing its entity-observation $\{o_{i,j}^{t+1}\} = [o_{i,1}^{t+1}, o_{i,2}^{t+1}, ..., o_{i,m}^{t+1}]$, last subtask $k_i^t$, history latent embedding $h_i^t$ produced by RNN.

## 3.1 Scalable Subtask Encoder

An agent's history, which comprises its observations and behavioral history, contains valuable information about its characteristics, such as velocity, attack distance, etc. Therefore, analyzing an agent's history can help us identify the most suitable subtask for the agent to perform. Typically [38], a recurrent neural network, such as LSTM [8] or GRU [3], is employed to construct the agent's history representation from its {action, observation} history. The history representation is then passed to a softmax operator, which samples a one-hot subtask for the agent. However, since actions and observations are task-dependent, this approach may lead to subtask encoder that overfits to the source task, making the source model non-generalizable. Considering that the subtasks history of an agent not only reflects its behavior history, but also is task-independent, and the entity-observations history is more general across tasks than observations history, our scalable subtask encoder assigns a subtask to each agent based on its {subtask, entity-observation} history rather than {action, observation} history, as illustrated in the **red solid-line box** in Figure 2.

The detail of the scalable subtask encoder is demonstrated in Figure 1 right. By replacing observations with entity-observations, we first design a structure that can obtain fixed-dimensional observation embeddings, regardless of the number of entity-observations contained in the observation. Additionally, since permuting the order of entity-observations in the observation does not change the information[13], we use a Permutation Invariant operator, namely, Gaussian Product, to obtain the observation embedding. Specifically, we first use a MLP parameterized by $\theta_e$ to embed the entity-observation as an entity-embedding:

$$(\mu_{i,j}^t, \sigma_{i,j}^t) = f_{\theta_e}(o_{i,j}^t), j = 1, ..., m \tag{2}$$

and the observation embedding $e_i^t$, which is also referred to as Env Cognition of agent $i$ in Figure 1, is constructed as:

$$e_i^t \sim \mathcal{N}(\mu_i^t, \sigma_i^t), \ where \ \mathcal{N}(\mu_i^t, \sigma_i^t) \propto \prod_{j=1}^{m} \mathcal{N}(\mu_{i,j}^t, \sigma_{i,j}^t) \tag{3}$$

Then we utilize a trajectory encoder based on GRU, which is parameterized by $\theta_h$, to obtain an agent's history representation. The GRU takes an agent's observation embedding $e_i^t$, last subtask $k_i^{t-1}$ and previous hidden state $h_i^{t-1}$ as inputs, and generates a new hidden state $h_i^t$ as its history representation:

$$h_i^t = f_{\theta_h}(e_i^t, k_i^{t-1}, h_i^{t-1}) \tag{4}$$

Afterwards, we use the Gumbel-Softmax trick with a categorical reparameterization [12] to assign a subtask $k_i^t$ to agent $i$ based on its current history representation $h_i^t$:

$$k_i^t \sim \text{Gumbel-Softmax}(h_i^t) \tag{5}$$

where $k_i^t \in \mathbb{R}^{n_k}$ is a $n_k$-dimensional one-hot vector and $n_k$ is a hyperparameter denoting the total number of subtasks. The Gumbel-Softmax operator allows our process of subtask assignment trainable by the way of gradient backpropagation.

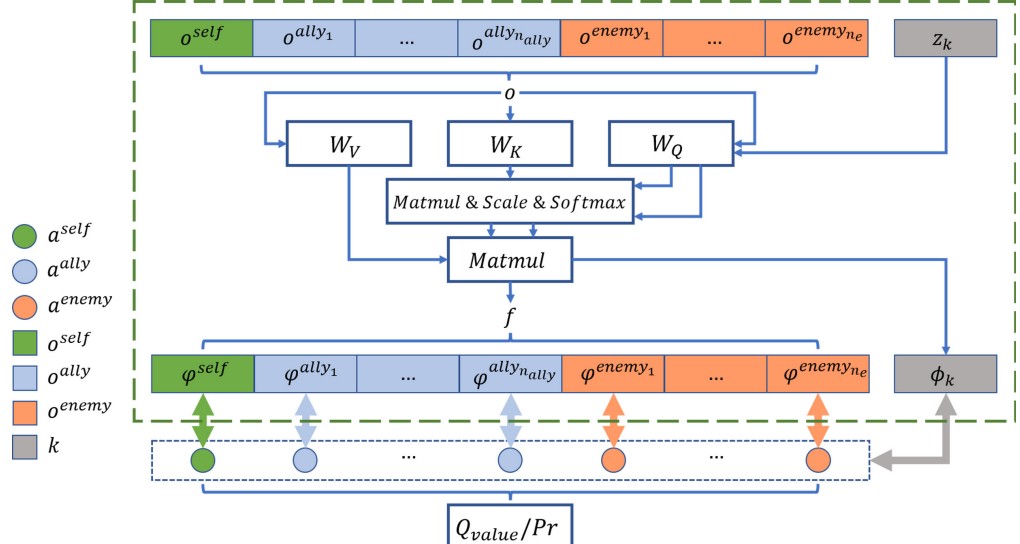

Figure 3: Adaptive Subtask Semantic module. The Adaptive Subtask Semantic Module endows subtasks and actions with adaptive semantics. $z_k$ denotes the embedding of subtask $k$. Entity-observations are denoted by $o = [o^{self}, o^{ally_1}, ..., o^{ally_{n_{ally}}}, o^{enemy_1}, ..., o^{enemy_{n_{enemy}}}]$ (we replace $n_{enemy}$ with $n_e$ in Figure 3 for convenience). With subtask embedding and entity-observations, we obtain the semantics $\phi_k$ of subtask $k$ and the semantics of actions $\varphi = [\varphi^{self}, \varphi^{ally_1}, ..., \varphi^{ally_{n_{ally}}}, \varphi^{enemy_1}, ..., \varphi^{enemy_{n_{enemy}}}]$ by utilizing the Attention mechanism [29].

## 3.2 Adaptive Subtask Semantics

The generalizable subtasks should maintain consistent yet scalable semantics across tasks. The traditional method of constructing subtask semantics is embedding one-hot subtasks into vector representations based on MLP. However, this approach endows target subtasks with completely identical semantics as source subtasks when deploying the source model to the target tasks, neglecting the difference between source and target tasks so as to restrict the zero-shot generalization of subtask semantics across tasks. To overcome this limitation, our adaptive subtask semantic module leverages the inductive bias that subtask semantics refers to the effects of an agent on entities when it performs a given subtask. This approach endows subtasks with semantics that can adaptively adjust based on the agent's effects on entities, thereby enabling greater zero-shot generalization of subtask semantics across tasks.

Since a subtask's semantics actually refers to the effects of an agent on entities when it performs this subtask, we construct subtask semantics as the weighted sum of entities, where the weight represents the degree of effects. And considering that we have obtained one-hot subtask and entity-observations, we utilize the Attention mechanism [29] without position embedding, which is scalable across tasks regardless of the number of entities and permutation invariant about the order of entity-observaitons in the observation, to obtain the subtask semantics, as illustrated in Figure 3. Specifically, we take one-hot subtask embedding

$$z_i^t = \text{Embedding}(k_i^t) \tag{6}$$

as the query and entity-observations' embedding as the keys as well as values: $\hat{Q}_i^t = W_Q z_i^t$, $K_i^t = W_K o_i^t$, $V_i^t = W_V o_i^t$, where $W_Q, W_K, W_V$ are learnable parameters and $o_i^t = [o_{i,1}^t, o_{i,2}^t, ..., o_{i,m}^t]$. Then we construct adaptive subtask semantics $\phi_i^t$ as follows:

$$\phi_i^t = \text{softmax}(\frac{\hat{Q}_i^t K_i^{t^T}}{\sqrt{d_K}})V_i^t, \quad \hat{Q}_i^t = W_Q z_i^t \tag{7}$$

where $d_K$ is the feature dimension of $K_i^t$. Additionally, to make the process of obtaining actions scalable across tasks, we extended the model structure of ASN [33]. Since the alignment between actions and entity-observations may discrepant from the prior, we construct the adaptive action semantics $\varphi_i^t = [\varphi_{i,1}^t, \varphi_{i,2}^t, ..., \varphi_{i,m}^t]$ by utilizing the Self-Attention mechanism taking the entity-observations as inputs:

$$\varphi_i^t = \text{softmax}(\frac{Q_i^t K_i^{t^T}}{\sqrt{d_K}})V_i^t, \quad Q_i^t = W_Q o_i^t \tag{8}$$

Subsequently, we calculate the value or probability of each action in current observation by comparing the similarity of subtask semantics with action semantics:

$$Q_{value}(a_j|o_i^t) \text{ or } Pr(a_j|o_i^t) = similarity(\phi_i^t, \varphi_{i,j}^t) \tag{9}$$

where $similarity$ is a trainable MLP taking the concatenate of $\phi_i^t$ and $\varphi_{i,j}^t$ as inputs.

# 4 Experiments

## 4.1 Experimental Setup

We evaluated the performance of DT2GS on the StarCraft Multi-Agent Challenge (SMAC) [22] and the multi-agent particle world environments (MPE) [16] (shown in Appendix D). SMAC contains several tasks that are similar but different, such as the *marine*-series tasks ({3m, 8m, 8m_vs_9m, 10m_vs_11m, 25m, 27m_vs_30m}) and the *stalker_zealot*-series tasks (2s3z, 3s5z, 3s5z_vs_3s6z). Besides, changing the number of agents and landmarks in MPE can also form a series of tasks. These tasks met our requirements of cross-task generalization, allowing us to evaluate the ability of DT2GS to generalize across different tasks.

In the generalization capability part of experiments, we selected ASN [33] and UPDeT [9] as baselines. ASN was chosen because it promotes the development of universal models across tasks in MARL. For the sake of fairness in comparison, we make ASN generalizable across tasks by utilizing the attention mechanism and use "ASN_G" to denote this generalizable ASN. UPDeT was selected because it constructs a universal model across tasks via policy decoupling with self-attention, which is also used in the adaptive subtask semantic module of DT2GS in Sec 3.2. In addition, we implemented DT2GS, UPDeT, and ASN_G based on MAPPO [40], which is considered SOTA in on-policy MARL.

In the asymptotic performance part of experiments, we added another five baselines for comparison: MAPPO[40], LDSA [38], ROMA [31], RODE [32], and HSD [37]. LDSA, ROMA, RODE, and HSD focus on concepts like skills/options/roles in MARL, which are similar to subtasks studied in our method.

Additionally, We set the number of subtasks $n_k$ to 4 and averaged all results over 4 random seeds. And the experiments are arranged as follows: Firstly, we evaluated the zero-shot generalization capability of DT2GS across tasks. Secondly, we analyzed the practical semantics of subtasks on the *marine*-series tasks. Thirdly, we conducted several experiments to exhibit the transferability of DT2GS across tasks. Finally, we demonstrated that DT2GS achieves SOTA performance on multi-task and most single-task problems in terms of asymptotic performance.

## 4.2 Zero-Shot Generalization across Tasks

In this section, we designed 8 different zero-shot generalization scenarios, where each scenario includes a target task that is more difficult than the source task. That is to say, the relationship from the source task to the target task is extrapolated. For example, the target task may be larger in scale (like 8m → 25m) or contain a greater disparity in military strength than the source task (3s_vs_4z → 3s_vs_5z). We deployed the model trained on the source tasks to the target tasks without any finetune. As shown in Figure 4, DT2GS significantly outperforms UPDeT and ASN_G in terms of zero-shot generalization capability, achieving an average test winning rate surpass of about 22% and 34%, respectively, over all 8 zero-shot generalization scenarios.

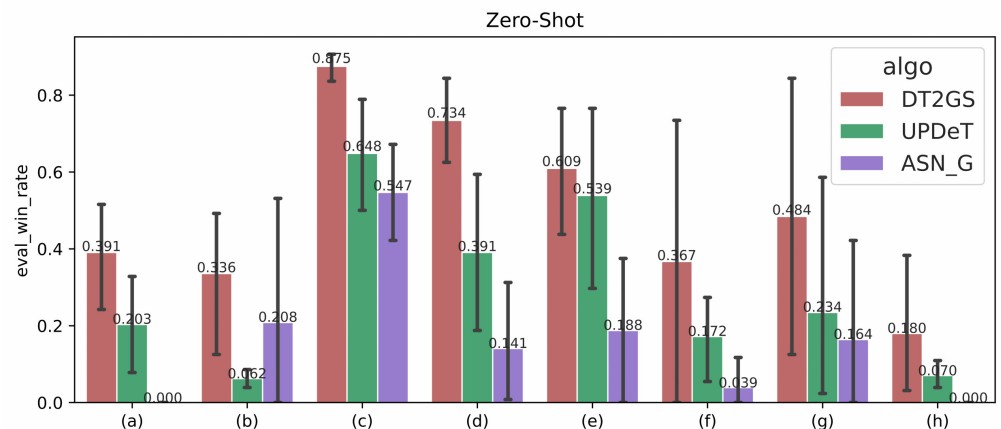

Figure 4: The figure shows a comparison of zero-shot generalization capability between DT2GS, UPDeT, and ASN_G across various source and target tasks. The horizontal axis represents the source task → the target task, where (a) 3s_vs_4z → 3s_vs_5z, (b) 2s3z → 3s5z, (c) 3s5z → 3s5z_vs_3s6z, (d) 8m → 8m_vs_9m, (e) 8m → 10m_vs_11m, (f) 8m → 25m, (g) 8m_vs_9m → 25m, and (h) 8m_vs_9m → 5m_vs_6m. The vertical axis represents the winning rate when deploying the source model to target task without any finetue. The red, green, and purple histograms correspond to DT2GS, UPDeT, and ASN_G, respectively. The missing histograms indicate that the winning rate of deploying the source model to target task is 0%.

## 4.3 Analysis of Subtask Semantics

In this section, we analyzed the practical semantics of subtasks based on the *marine*-series tasks. We first trained a DT2GS model from scratch on the 8m task and then deployed it to 8m and 10m_vs_11m, as shown in Figure 5 left. In the 8m and 10m_vs_11m tasks, the DT2GS agents initially utilized subtask_1 to scatter the formation and advance towards the enemy at $t = 1$. When the enemies reached the attack range of DT2GS agents at $t = 6$, most of these agents selected subtask_2 to focus fire on enemies while also paying attention to their own health in order to move back when in danger, thereby avoiding the decrease in firepower caused by personnel reduction. And other agents chose subtask_4 to slightly adjust their relative position with the enemies to seek a better position for firepower output. At $t = 13$, some agents sensed that their allies' health was too low while their own health was safe enough. These agents switched subtask from subtask_2 to subtask_3, which means charging. And the specific behavior was to attack while moving closer to the enemies' position to attract hatred so as to avoid the decrease in firepower caused by the death of allies. At the end of the battle (after $t = 21$), all agents changed their subtasks to subtask_3 to concentrate their firepower to end the battle. We observed that under the subtask of charge, even if an agent was close to death, it would not move back to disperse hatred as overall withdrawal would lead to failed combat.

Furthermore, we conducted zero-shot generalization experiments on additional *marine* tasks, and the corresponding change process of subtask percentage is shown in Figure 6. We observed that the change process of each subtask's percentage in the source task (8m) is quite similar to that in the target tasks (10m_vs_11m, 5m_vs_6m, 8m_vs_9m, 25m, and 27m_vs_30m). This result demonstrates that the subtasks decomposed by DT2GS from the source task (8m) are task-independent across tasks and

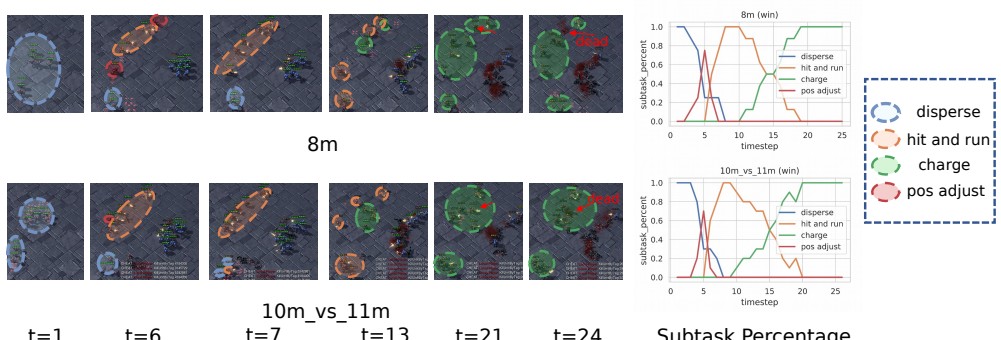

Figure 5: Visualization of the practical semantics of subtasks (left) and the change process of subtask percentage (right) in 8m (up) and 10m_vs_11m (down). The results for both tasks are obtained from the source model's evaluation on the corresponding task, where the source model is acquired by learning from scratch in task 8m.

therefore effective for the target tasks. Both results from Figure 5 and Figure 6 further suggest that DT2GS policy has sound zero-shot generalization capability by decomposing the task into a series of task-independent subtasks.

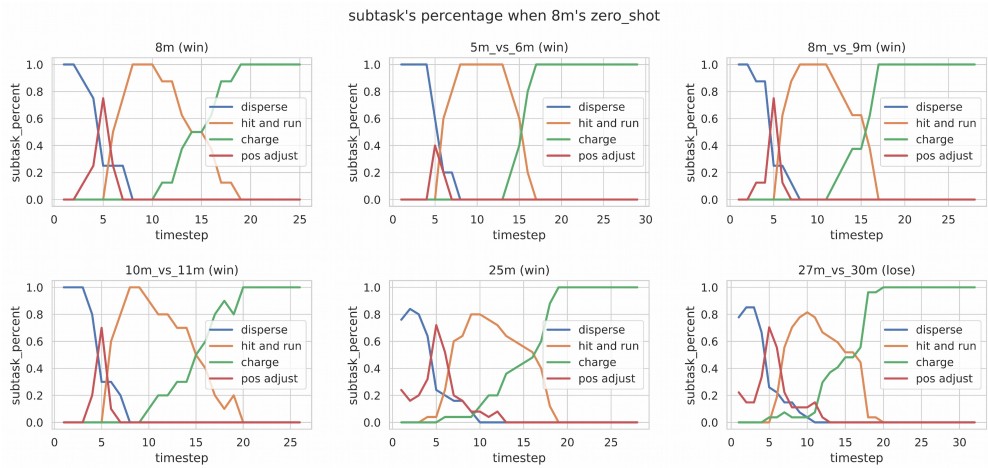

Figure 6: Changes of subtask percentage when deploying the source model, which is learned from scratch on 8m, to 6 $marine$ tasks. The subtask percentage is calculated as $p(k) = \frac{N_k}{N_a}$, where $p(k)$ is the percentage of subtask $k$, $N_k$ is the number of agents that select subtask $k$, and $N_a$ is the total number of agents.

### 4.4 Transferability across Tasks

Figure 7 (a) illustrates four scenarios that we designed to evaluate the transferability of DT2GS policy. In all four scenarios, the relationships from the source task to the target task are extrapolation, including an increase in entity types (3s5z_vs_3s6z → 1c3s5z), an increase in the number of enemies (3s5z → 3s5z_vs_3s6z and 3s_vs_4z → 3s_vs_5z), as well as an increase in the total scale of entities (8m_vs_9m → 27m_vs_30m).

As we can see, DT2GS_finetune demonstrated significantly efficient convergence as well as higher and more stable asymptotic performance compared with UPDeT_finetune and ASN_G_finetune in all transfer scenarios. Notably, in the scenario where the source task was 3s5z (easy) and the target task was 3s5z_vs_3s6z (superhard), DT2GS_finetune achieved optimal performance with only 6400 steps interaction with environment, whereas learning from scratch required at least 6 million steps. Furthermore, compared to other baselines of learning from scratch, DT2GS_finetune accelerated

the model's convergence on target tasks by an average of $100\times$. In conclusion, DT2GS_finetune exhibited better performance than baselines on the following evaluation metrics [28, 4]: jumpstart, time to threshold and asymptotic performance, demonstrating significantly sufficient transferability.

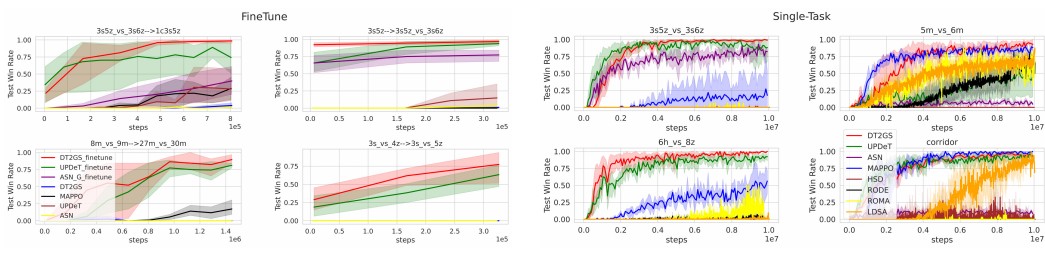

(a) Transferability           (b) Single-Task

Figure 7: (a) Comparison of transferability between DT2GS and baselines, where baselines include transfer of UPDeT (UPDeT_finetune), ASN_G (ASN_G_finetune) and learning from scratch of DT2GS, UPDeT, ASN, MAPPO. (b) Comparison of performance on Single-Task between DT2GS and baselines, including UPDeT, ASN, MAPPO, HSD, RODE, ROMA and LDSA, on 3 superhard tasks (3s5z_vs_3s6z, 6h_vs_8z, corridor) and 1 hard task (5m_vs_6m).

## 4.5 Performance on Multi-task and Single-task

In this section, we first designed 2 multi-task problems, including the *marine*-series tasks ({3m, 8m, 8m_vs_9m, 10m_vs_11m}) and *stalker_zealot*-series tasks ({2s3z, 3s5z, 3s5z_vs_3s6z}), to demonstrate the representational capacity of DT2GS. In each multi-task problem, the policy interacted synchronously with multiple tasks that make up this multi-task to collect data, which was shuffled and used to update the policy. As shown in Figure 8, DT2GS exhibited better learning efficiency and more stable asymptotic performance compared with UPDeT and ASN_G.

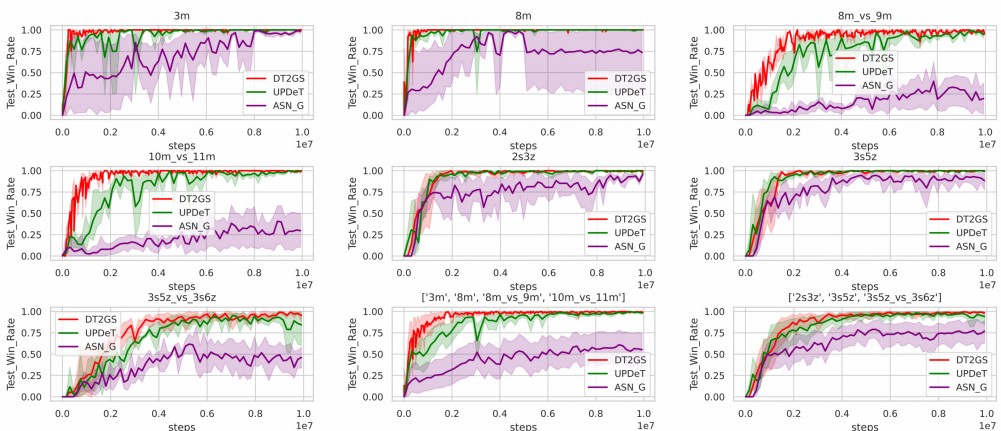

Figure 8: Comparison of performance on multi-task problems between DT2GS and baselines, including UPDeT and ASN_G.

Subsequently, we also compared DT2GS with baselines on single-task scenarios. We selected four representative scenarios, including superhard / hard tasks, as shown in Figure 7 (b), while performance on other scenarios is presented in the Appendix B. Our evaluation metrics included learning efficiency and final asymptotic performance. Compared to MAPPO, DT2GS significantly improved both metrics, particularly in superhard tasks such as 3s5z_vs_3s6z and 6h_vs_8z. In addition, compared to other baselines based on MAPPO, including UPDeT and ASN, DT2GS consistently outperformed them in terms of learning efficiency and asymptotic performance, on average.

# 5 Conclusion

Model generalization has emerged as a promising approach to reduce training costs. In this paper, we proposed DT2GS, an effective approach for addressing the problem of model generalization across tasks in the MARL field. Our insight is that task-independent subtasks exist across tasks, making it possible to generalize the model across tasks. Based on this insight, we assumed that the model can be generalized to target tasks if it can decompose task-independent subtasks from source tasks. The challenge then becomes ensuring that the subtasks we decompose from the source task are truly task-independent. Regarding this issue, we proposed two properties that enable task-independence of subtasks: (1) avoiding overfitting to the source task, (2) maintaining consistent yet scalable semantics across tasks. Then we proposed DT2GS to endow the subtasks with these two properties by introducing the scalable subtask encoder and the adaptive subtask semantic module, respectively. Empirical results demonstrated that DT2GS can decompose tasks into a series of generalizable subtasks, leading to a generalizable MARL policy. Nevertheless, it would be beneficial to consider task-specific subtasks as well when there is a significant distribution shift between source and target tasks. In our future work, we will focus on expanding the generalization scenarios of DT2GS to address this limitation.

# 6 Acknowledgements

This work is partially supported by the NSF of China(under Grants 61925208, U22A2028, 62222214, 62002338, 62102399, U19B2019, 92364202), CAS Project for Young Scientists in Basic Research(YSBR-029), Youth Innovation Promotion Association CAS and Xplore Prize.

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

# A   Related Work

**Transfer learning across tasks in online MARL**     Transfer learning across tasks is a promising approach to accelerate the convergence of models, especially in large-scale MAS. The first method of transfer learning in online MARL domains is BITER [2], which uses the joint policy learned in the source task to bias the initial policy of agents in the target tasks. After more than a decade of development, some methods have been proposed in this field. In this regard, we classify these methods into two categories based on their characteristics, namely Network-design-based methods and Task-embedding-based methods.

The Network-design-based methods implicitly reuse knowledge from source tasks by constructing universal model structure across tasks using Population Invariant Structure such as Transformer [9] or GNN [1]. Since the existence of varying state/observation/action space across tasks, the Network-design-based methods are the foundation of all other methods. And the Task-embedding-based methods are knowledge transfer by calculating task similarity using learned task embeddings that capture task dynamics [2, 5, 14, 23, 19]. For example, MATTAR [19] models the target task representation as a weighted sum of a series of source task representations and decodes the task representation to the policy weight that is suitable for the task by utilizing a hypernetwork. And MATE [23] models task representation by reconstructing the task dynamics, so as to achieve task differentiation and fast adaptation of policy on tasks.

Previous works focused on multi-agent transfer learning based on network-design or task-embedding, but lacked of efficient knowledge reuse, leading to limited generalization capability. Our work (DT2GS) leverages knowledge reuse by ensuring semantic consistency and scalability between diverse tasks and preventing over-fitting of source tasks, which greatly improves transferability and even zero-shot generalization capability.

**Transfer learning across tasks in other paradigms**     There are some methods dedicated to transfer learning across tasks in MARL belonging to other paradigms. For example, ODIS [41], belonging to the offline RL paradigm, discovers generalizable skills from multi-task offline data to enhance the model's generalization capability to unseen tasks. Besides, EPC [15] and DyMA-CL [34] fulfill knowledge transfer across tasks by way of Curriculum Learning.

**Skills/Options/Roles/Subtasks in MARL**     There are some works in MARL focusing on concepts like skills/options/roles, which are similar to subtasks studied in our method. LDSA [38] learns dynamic subtask assignments and constructs different policies for different subtasks, so enabling agents to possess specific abilities to handle different subtasks. ROMA [31] introduces the role concept into MARL in the form of Gaussian distribution and assigns certain subtasks to agents with similar roles. RODE [32] further studies how to efficiently discover the roles set by decomposing joint action spaces into restricted role action spaces. HSD [37] is proposed to discover complementary and useful skills for cooperative team play in MARL.

**ASN [33] && UPDeT [9]**     For model generalization in MAS, a generalizable model structure is necessary for addressing the problem of varying state/observation/action space ($\mathcal{S}/\mathcal{O}/\mathcal{A}$) across tasks. Here we define the agents controlled by MARL policy and agents built-in tasks as entities. As demonstrated in ASN [33], an agent's observation $o_i$ can be constructed as a concatenation of $m$ entity-observations: $o_i = [o_{i,1}, o_{i,2}, ..., o_{i,m}]$, where $o_{i,1}$ is the observation of agent $i$ on itself and environment, and the rest are the observations of agent $i$ on other $m-1$ entities. Additionally, action space $\mathcal{A}$ can be decomposed into two components: $\mathcal{A}^{self}$, which consists of actions affecting the agent itself or the environment, and $\mathcal{A}^{interactive}$, which contains actions that directly interact with other entities. This alignment between entity-observations and actions, which is referred to as action semantics, forms the foundation for computing the value or probability of an action based on its aligning entity-observation, leading to a generalizable model structure across tasks.

Thereafter, by combining ASN with Transformer [29], UPDeT [9] develops a type of population-invariant network (PIN) to further improve the model's generalization capability. Specifically, entity-observations are taken as queries, keys, and values to derive an attention output, as shown in Formula (10) and Formula (11).

$$Q_i^t = W_Q[o_{i,1}^t, o_{i,2}^t, ..., o_{i,m}^t],\ K_i^t = W_K[o_{i,1}^t, o_{i,2}^t, ..., o_{i,m}^t],\ V_i^t = W_V[o_{i,1}^t, o_{i,2}^t, ..., o_{i,m}^t] \quad (10)$$

$$\varphi_i^t = [\varphi_{i,1}^t, ..., \varphi_{i,m}^t] = \text{softmax}(\frac{Q_i^t K_i^{t^T}}{\sqrt{d_K}})V_i^t \qquad (11)$$

Subsequently, the attention output $\varphi_{i,1}^t$ and $\varphi_{i,j}^t, j = 2, ..., m$ are utilized to obtain the Q-value or probability of actions in $\mathcal{A}^{self}$ and $\mathcal{A}^{interactive}$, respectively. As shown in ODIS [41], UPDeT [9] and [1], the pipeline of PIN has been regarded as a foundational framework for the model's generalization across tasks in MARL. Therefore, the PIN is also employed in DT2GS.

# B  Asymptotic Performance on Single-Task problems

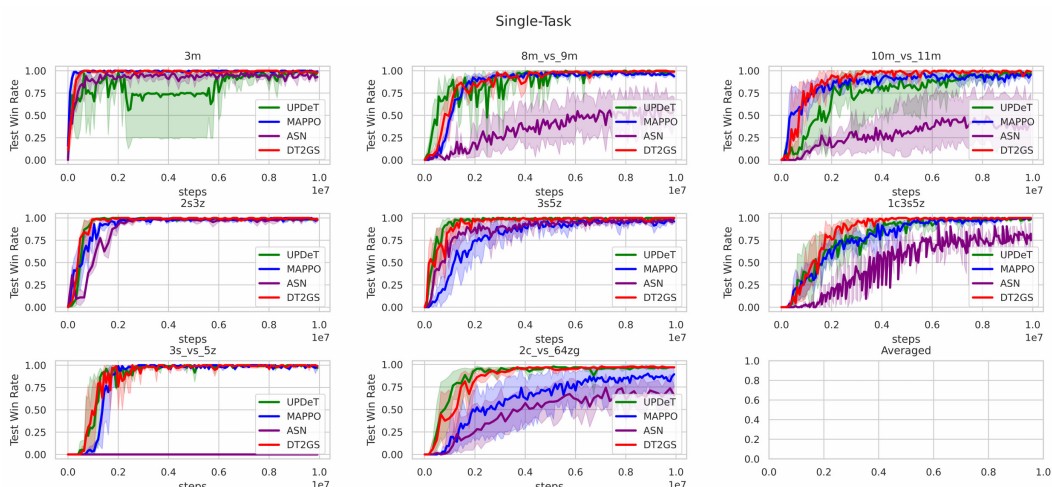

Figure 9: Comparison of DT2GS performance with baselines, including UPDeT, ASN, and MAPPO, on Single-Task Problems.

# C   Consistent yet Scalable semantics of subtask

DT2GS endows subtasks with consistent yet scalable semantics across tasks. **For instance, the semantics of subtask $k$ could involve tempting 2 enemies in the source task while 4 enemies in the target task. The term "consistency" refers to $k$ tempting enemies in both the source and target tasks, while "scalability" refers to the changed impact on entities**. Figure 10 illustrates the subtask semantics, which are represented by the subtask's attention on all entities. We observed that DT2GS can adapt subtask semantics to different task contexts effectively, reflecting its scalability on subtask semantics across tasks. In our experiments, agents paid attention solely to allies as they employed the subtask "disperse" to line up during the t_1 period. During the t_2 period, agents initially paid attention on both allies and enemies as they utilized the subtask "hit and run" to focus fire on enemies while monitoring their own health and allies' position. Later in t_2, agents shifted their focus to enemies as they employed the subtask "charge" to concentrate their firepower to end the battle. The last period, t_3, signifies the agents' demise.

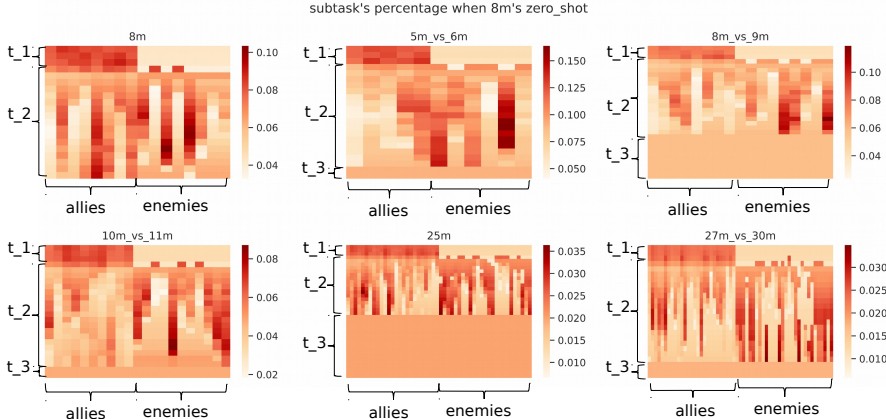

Figure 10: Visualization of the subtask semantics, which represented by the subtask's attention on all entities. We trained a model on the source task 8m and evaluated its performance on 6 different *marine* tasks. To visualize the semantics of subtasks assigned to the agent during an episode, we plotted the subtask attention on all entities over time. The resulting visualization is shown above, where each cell represents the attention weight of a subtask-entity pair at a specific timestep. The horizontal axis denotes the entities present in the task, ordered from allies to enemies from left to right, while the vertical axis represents the chronological order of timesteps in the episode from top to bottom.

## D Generalization on MPE

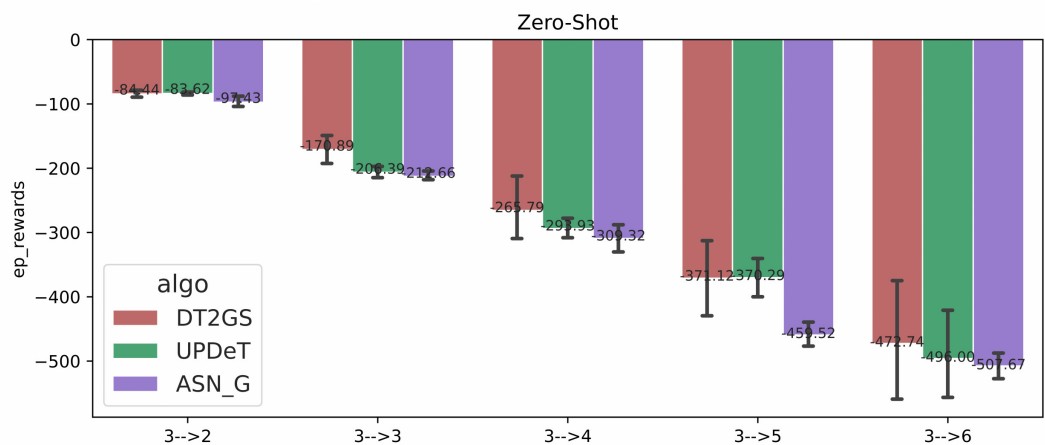

Figure 11: The figure shows a comparison of zero-shot generalization capability between DT2GS, UPDeT, and ASN_G on the physical deception task (Spread) in the multi-agent particle world environments (MPE) [16]. The horizontal axis represents the source task → the target task. For example, "3→4" indicates that the source task is set by 3 agents and 3 landmarks, while the target task is set by 4 agents and 4 landmarks. The vertical axis represents the average episode reward when deploying the source model to the target task without any finetue. The red, green, and purple histograms correspond to DT2GS, UPDeT, and ASN_G, respectively. As we can see, DT2GS significantly outperforms UPDeT and ASN_G in terms of zero-shot generalization capability, achieving an average episode reward surpass of about 17.05 and 44.32, respectively

## E Ablation Study

To investigate the effect of the number of subtasks $n_k$ on the model's zero-shot generalization capability, we conducted an ablation study, as shown in Figure 12. Our results indicate that appropriately increasing the number of subtasks can improve the model's zero-shot generalization capability, but setting $n_k = 6$ leads to degradation due to the limited number of generalizable subtasks across tasks. Additionally, we found that increasing $n_k$ from 4 to 5 does not significantly improve performance, particularly in the zero-shot generalization scenario 8m → 25m. To balance efficiency and effectiveness, we set $n_k = 4$ for all experiments presented in this paper.

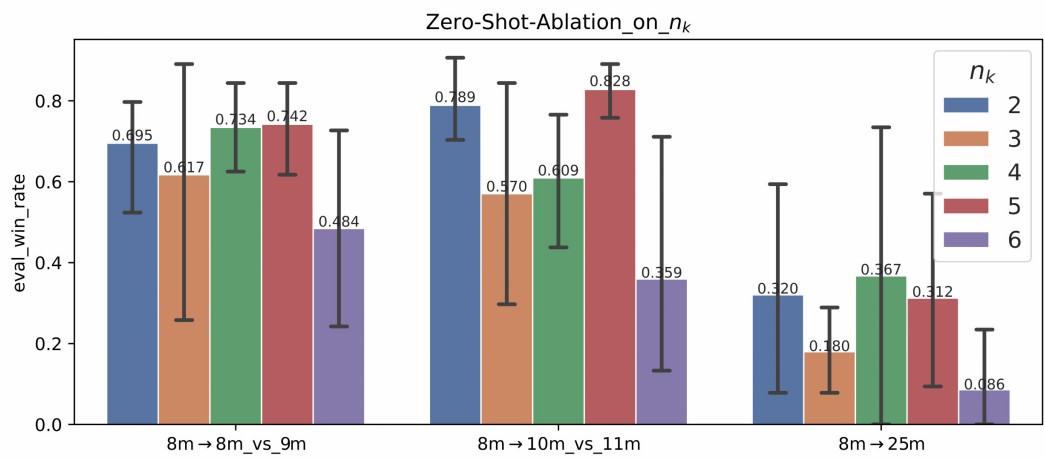

Figure 12: Ablation study on the impact of subtask number $n_k$ on the model's zero-shot generalization capability.

# F   List of symbol definitions

Table 1: List of symbol definitions

| Symbol | Definitions |
|--------|-------------|
| $o_i$ | the observation of agent $i$ |
| $o_{i,j}$ | the entity-observation of agent $i$ on entity $j$ |
| $\{o_{i,j}^t\}$ | all entity-observations of agent at time $t$ |
| $o^{self}$ | the entity-observation of agent on itself and environment |
| $o^{ally_j}$ | the entity-observation of agent on its $j$th ally |
| $o^{enemy_j}$ | the entity-observation of agent on its $j$th enemy |
| $a_i^t$ | the action of agent $i$ at time $t$ |
| $k_i^t$ | the subtask assigned to agent $i$ at time $t$ |
| $h_i^t$ | the history latent embedding of agent $i$ at time $t$ |
| $\mathcal{N}(\mu_{i,j}^t, \sigma_{i,j}^t)$ | the embedding Gaussian Distribution of entity-observation of agent $i$ on entity $j$ at time $t$ |
| $\mathcal{N}(\mu_i^t, \sigma_i^t)$ | the embedding Gaussian Distribution of observation of agent $i$ at time $t$ |
| $e_i^t$ | the observation embedding of agent $i$ at time $t$ |
| $\phi_i^t$ | the adaptive semantics of subtask assigned to agent $i$ at time $t$ |
| $\varphi_i^t$ | the semantics of all actions for agent $i$ at time $t$ |

# G   Hyperparameters

Table 2: List of Hyperparameters

| Hyperparameters | Value | Algorithms |
|-----------------|-------|------------|
| hidden layer dimension of DT2GS's Encoder | 8 | DT2GS |
| MLP's hidden layer dimension | 64 | DT2GS, UPDeT, ASN, ASN_G, MAPPO |
| attention's hidden layer dimension | 64 | DT2GS, UPDeT, ASN_G |
| attention's heads | 3 | DT2GS, UPDeT, ASN_G |
| number of subtasks | 4 | DT2GS |
| optimizer | Adam | DT2GS, UPDeT, ASN, ASN_G, MAPPO |
| learning rate of actor and critic | 0.0005 | DT2GS, UPDeT, ASN, ASN_G, MAPPO |

To ensure consistency and comparability across our experiments, we aimed to share hyperparameters among algorithms wherever possible. We provide a list of the shared hyperparameters in Table 2, where the "Algorithms" column indicates which algorithms have corresponding hyperparameters in common.

## H Pseudocode

---

**Algorithm 1** DT2GS based on MAPPO

---

**Input:** The parameters $\theta_{En}^{\pi}$ for Scalable Subtask Encoder of actor $\pi$; the parameters $\theta_{De}^{\pi}$ for Adaptive Action Decoder of actor $\pi$; the parameters $\theta^V$ for critic $V$; the number $n_k$ for subtasks

**Output:** The Scalable Subtask Encoder $\pi_{En}$ and Adaptive Action Decoder $\pi_{De}$ for actor $\pi$; the critic $V$

1: Initialize $\theta_{En}^{\pi}, \theta_{De}^{\pi}, \theta^V, n_k$
2: Initialize the total timesteps $step_{\max}$ interaction with environment; the total timesteps $T$ of an episode; the number of episodes $batch\_size$ for each actor/critic update
3: Initialize $step \leftarrow 0$
4: **while** $step \leq step_{\max}$ **do**
5:     set data buffer D = { }
6:     **for** $idx = 1$ **to** $batch\_size$ **do**
7:         $\tau = []$ empty list
8:         initialize actor RNN states $h_{1,\pi}^0, ..., h_{n,\pi}^0$ for each agent
9:         initialize critic RNN states $h_{1,V}^0, ..., h_{n,V}^0$ for each agent
10:        initialize subtasks $k_1^0, ..., k_n^0$ for each agent   ▷ Initialize subtasks with $n_k$-dim zero vector
11:        **for** timestep $t = 1$ **to** $T$ **do**
12:           **for all** agents $i$ **do**
13:              $k_i^t, h_{i,\pi}^t = \pi_{En}(o_i^t, k_i^{t-1}, h_{i,\pi}^{t-1}; \theta_{En}^{\pi})$   ▷ Call for Scalable Subtask Encoder Module 2
14:              $a_i^t = \pi_{De}(o_i^t, k_i^t; \theta_{De}^{\pi})$            ▷ Call for Adaptive Action Decoder Module 3
15:              $v_i^t, h_{i,V}^t = V(s_i^t, h_{i,V}^{t-1}; \theta^V)$
16:           **end for**
17:          Execute actions $\boldsymbol{a^t}$, observe $r^t, s^{t+1}, \boldsymbol{o^{t+1}}$
18:          $\tau += [s^t, \boldsymbol{o^t}, \boldsymbol{h_{\pi}^t}, \boldsymbol{h_V^t}, \boldsymbol{k^t}, \boldsymbol{a^t}, r^t, s^{t+1}, \boldsymbol{o^{t+1}}]$
19:        **end for**
20:        $step += T$
21:        Compute advantage estimate $\hat{A}$ via GAE on $\tau$
22:        Compute reward-to-go $\hat{R}$ on $\tau$
23:        Split trajectory $\tau$ into chunks of length L
24:        **for** l=0,1,...,T//L **do**
25:          $D = D \cup (\tau[l : l + T], \hat{A}[l : l + L], \hat{R}[l : l + L])$
26:        **end for**
27:     **end for**
28:     **for** mini-batch $k = 1, ..., K$ **do**
29:       $b \leftarrow$ random mini-batch from $D$ with all agent data
30:       **for** each data chunk $c$ in the mini-batch $b$ **do**
31:         update RNN hidden states for $\pi_{En}$ and $V$ from first hidden state in data chunk
32:       **end for**
33:     **end for**
34:     Adam update $\theta_{En}^{\pi}, \theta_{De}^{\pi}$ on actor loss with data $b$
35:     Adam update $\theta^V$ on critic loss with data $b$
36: **end while**
37: **Return** $\pi_{En}, \pi_{De}, V$

---

---

**Algorithm 2** Scalable Subtask Encoder

---

**Input:** The parameters $\theta_{En}^{\pi}$ for Scalable Subtask Encoder of actor $\pi$, agent $i$'s observation $o_i^t$ in timestep $t$, agent $i$'s subtask $k_i^{t-1}$ and actor RNN state $h_{i,\pi}^{t-1}$ in timestep $t-1$

**Output:** agent $i$'s subtask $k_i^t$ and actor RNN state $h_{i,\pi}^t$ in timestep $t$

1: Compute entity-embedding $(\mu_{i,j}^t, \sigma_{i,j}^t), j = 1, ..., m$ by Formula (2)
2: Compute observation-embedding $e_i^t$ by Formula (3)
3: Compute actor RNN state $h_{i,\pi}^t$ by Formula (4)
4: Compute subtask $k_i^t$ with $k_i^t \sim$ Gumbel-Softmax$(h_{i,\pi}^t)$
5: **Return** $k_i^t, h_{i,\pi}^t$

---

---

**Algorithm 3** Adaptive Action Decoder

---

**Input:** The parameters $\theta_{De}^{\pi}$ for Adaptive Action Decoder of actor $\pi$, agent $i$'s observation $o_i^t$ and subtask $k_i^t$ in timestep $t$,

**Output:** agent $i$'s action $a_i^t$ in timestep $t$

1: Compute subtask embedding $z_i^t$ with $z_i^t =$ Embedding$(k_i^t)$
2: Compute adaptive subtask semantics $\phi_i^t$ by Formula (7)
3: Compute action semantics $\varphi_i^t$ by Formula (8)
4: Sample action $a_i^t$ from the actions' probability distribution $Pr_i^t$ computed by Formula (9)
5: **Return** $a_i^t$

---

