# OpenReview forum: "Decompose a Task into Generalizable Subtasks in Multi-Agent Reinforcement Learning"
_NeurIPS.cc/2023/Conference — NeurIPS 2023 poster_

### Official Review · Reviewer_NoDQ · 2023-06-21

**Soundness:** 2 fair
**Presentation:** 2 fair
**Contribution:** 2 fair
**Rating:** 4
**Confidence:** 3

**Summary:**

The paper proposes a new neural network architecture for representing agent policies in multi-agent reinforcement learning. The architecture consists of two parts: (i) a subtask encoder that chooses the subtask to perform based on the subtask-observation history and (ii) a subtask semantics module that chooses the action based on the current observation and the output of the subtask encoder. The user specifies the number of subtasks as a hyperparameter and both modules are trained end-to-end. Experimental evaluation shows that this architecture enables SOTA performance in single-task as well as multi-task settings. Furthermore, the approach enables training policies that generalize to new scenarios in zero-shot and finetuning settings. A qualitative analysis shows that the subtasks learned can be meaningful and interpretable.

**Strengths:**

1. The idea of splitting the policy into two modules to decompose the overall task into subtasks is interesting and seems to work pretty well in the StarCraft environment. The ability to discover interpretable subtasks is an important benefit of the approach.
2. Zero-shot generalization abilities are improved by this architecture. The learned policy generalizes reasonably well to new scenarios obtained by modifying the original task.
3. The overall architecture is logical and appears easy to implement.

**Weaknesses:**

1. The work seems somewhat incremental and I am unsure if there is sufficient reason to believe the generality of the approach w.r.t. different MARL applications and scenarios. The results are purely empirical and it would strengthen the paper to consider another environment different from the StarCraft domain studied in this paper.
2. Some design choices are not fully justified. For example, hiding the action history from the subtask encoder is mentioned as a reason to enable cross-task generalization. However, the observation history might contain information about the action history in some environments and it is unclear if hiding the action history in such a case will make a difference. In fact, I believe it is important to include an ablation in which the action history is also given to the subtask encoder to show that the proposed method is better.
3. The architecture seems to use standard techniques and it would really improve the paper to explain the difference w.r.t. prior work and what is novel about this particular architecture---e.g., how it differs from the baselines and what parts of the architecture are borrowed from prior work.
4. The decomposition of the task into subtasks can also be achieved using a hierarchical approach such as the one in [1]. I believe that it is important to compare the empirical performance of the approach presented in this paper with that of a hierarchical approach.

[1] Yang, Jiachen, Igor Borovikov, and Hongyuan Zha. "Hierarchical Cooperative Multi-Agent Reinforcement Learning with Skill Discovery." International Conference on Autonomous Agents and Multi-Agent Systems. 2020.

**Questions:**

1. What parts of the architecture are novel and how does it differ from prior work?
2. Is there any prior work that tries to decompose the policy into a subtask encoder and a subtask semantics module, either in the cooperative multi-agent setting or the single-agent setting?
3. Would it make sense to consider freezing the subtask semantics module (after training on multiple tasks) during finetuning to perform new tasks that can be accomplished using the same subtasks? This might demonstrate that the learned subtasks generalize to completely new tasks which may require performing the subtasks in a different order.
4. Is the architecture trainable in a hierarchical way---i.e., train the subtask encoder at a higher level (grouping transitions in some way) than the subtask semantics module?

**Limitations:**

Limitations are not explicitly discussed and it would be good to include them---e.g., reliance on the assumption that the action space can be decomposed in a certain way (self-action and actions affecting other agents).

---

> ### Author Rebuttal · Authors · 2023-08-10
>
> We greatly appreciate your advice on further enhancing this paper. We would like to discuss them one by one and would greatly appreciate any further discussion on the matter.
>
> 1. > **Weakness 1**: Experiments on another environment are needed
>
>    We conducted zero-shot experiments on the physical deception task (Spread) in the multi-agent particle world environments (MPE). The experimental results can be seen in "**Enhancement 2**" of our global response. It can be seen that DT2GS outperforms UPDeT and ASN_G in terms of zero-shot generalization capability, achieving an average episode reward surpass of about 17.05 and 44.32, respectively.
>
> 2. > **Weakness 2**: It's unclear whether hiding the action history can enable cross-task generalization or not. An ablation is needed.
>
>    Perhaps the reviewer misunderstood our approach. Actually, rather than just hiding action history, replacing action history with subtask history is one of the reasons for achieving cross-task generalization. Besides, since different tasks have different action spaces, giving the action history to the subtask encoder will also make DT2GS non-generalizable in terms of model structure. And replacing action history with subtask history not only preserves the agent's abstract (high-level) behavior history but also enables DT2GS generalizable across tasks in terms of model structure.
>
> 3. > **Weakness 3 && Question 1**: Which parts of DT2GS are borrowed from prior work and which parts are new?
>
>    To adapt the model structurally to varying observation/action dimensions, DT2GS adopted the population-invariant network (PIN) structure developed in UPDeT, where the proposal of PIN was also promoted by the network design of ASN.
>
>    The innovations in our model architecture mainly include two parts:
>
>    (a) Designed a subtask encoder from the perspective of cross-task generalization by a structure-designed approach.
>
>    (b) Constructed an Adaptive Subtask Semantics module, which was not proposed by any previous work, endowing subtasks with consistent yet scalable semantics across tasks to improve model's generalization capability.
>
>
>
> 4. > **Weakness 4**: Lack of empirical comparison with HSD.
>
>    HSD is proposed to discover complementary skills (subtasks) in MARL, so it is closely related to DT2GS. Therefore, we added HSD[1] into our reference and as one of the baselines in our experiments. The specific results can be seen in "**Enhancement 1**" of our global response. It can be observed that DT2GS significantly outperforms HSD in terms of asymptotic performance.
>
> 5. > **Question 2**: Is there any prior work that tries to decompose the policy into a subtask encoder and a subtask semantics module?
>
>     DT2GS is the first work that endows subtasks with consistent yet scalable semantics across tasks by the Adaptive Subtask Semantics module. There are several works containing a subtask encoder in its model to construct subtasks/skills/roles/options [1,2,3,4,5]. However, apart from ODIS[1], none of these works designed a subtask encoder from the perspective of cross-task generalization. And different from ODIS which endows the subtask encoder with generalization by a data-driven approach, DT2GS endows the subtask encoder with generalization by a structure-designed approach.
>
>    > [1] Zhang et al., "Discovering generalizable multi-agent coordination skills from multi-task offline data", ICLR 2022.
>    >
>    > [2] Yang et al., "LDSA: Learning dynamic subtask assignment in cooperative multi-agent reinforcement learning", NeurIPS 2022.
>    >
>    > [3] Wang et al., "RODE: Learning roles to decompose multi-age", ICLR 2021.
>    >
>    > [4] Wang et al., "Roma: Multi-agent reinforcement learning with emergent", ICML 2020.
>    >
>    > [5] Yang et al., "Hierarchical cooperative multi-agent reinforcement learning with skill discovery", AAMAS 2020.
>
>
>
> 6. > **Question 3**: Conduct transfer under the setup of freezing the subtask encoder.
>
>    Actually, the zero-shot generalization experiment we conducted froze both the subtask encoder and subtask semantics module, which is more stringent than the experimental setup of just freezing the subtask encoder. Therefore, the zero-shot generalization experiments have demonstrated that the learned subtasks generalize well to completely new tasks which may require performing the subtasks in a different order.
>
> 7. > **Question 4**: Is DT2GS trainable in a hierarchical way?
>
>    Yes, we can train the subtask encoder at a higher level than the subtask semantics module. But hierarchically training the model is more commonly used in the setup of sparse reward. With intrinsic rewards, such training way is helpful for temporal credit assignments.
>
> 8. > **Limitations**: Assumption that the action space can be decomposed in a certain way
>
>    We added this limitation in our revised paper. In the future, we will improve DT2GS in ways like automatically decomposing the action space to alleviate this limitation.

---

> > ### Author Response · Authors · 2023-08-18
> > **Sincerely invite for further discussion.**
> >
> > Dear reviewer, as the discussion stage is coming to a close, we kindly await your comments and suggestions. We hope that our responses and additional experiments conducted during the rebuttal period have addressed all your questions or concerns. Do you have any further concerns or suggestions you would like to raise? We are eager to engage in a productive conversation with you.

---

> > > ### Comment · Reviewer_NoDQ · 2023-08-19
> > >
> > > I thank the authors for the detailed response. The additional experiments and the rebuttal addressed many of my concerns and hence I am revising my score. I am still unsure about the technical novelty of the approach but I wouldn't mind if the paper is accepted.

---

> > > > ### Author Response · Authors · 2023-08-19
> > > > **Thank you for your reply!**
> > > >
> > > > > **About the technical novelty of the approach**
> > > >
> > > > The main contribution of our work is that, through the Scalable Subtask Encoder and the Adaptive Subtask Semantics modules we designed, DT2GS is endowed with sufficient generalization capability to generalize the task-independent subtasks it decomposed from the source task to target tasks. Furthermore, the experiments illustrated in Figure 5 and Figure 6 demonstrate the task-independency of the subtasks decomposed by DT2GS, providing evidence for our contribution. Moreover, our design of the Scalable Subtask Encoder and the Adaptive Subtask Semantics modules is novel and has not been proposed in previous works. Specifically,
> > > >
> > > > - Adaptive Subtask Semantics: DT2GS is the first work that proposes the Adaptive Subtask Semantics module to endows subtasks with consistent yet scalable semantics across tasks to improve model's generalization capability.
> > > > - Scalable Subtask Encoder: Different from previous work like ODIS [1] which endows the subtask encoder with generalization by a data-driven approach, DT2GS endows the subtask encoder with generalization by a structure-designed approach. Additionally, this structure-designed approach can also be combined with the data-driven approach to further improve model's generalization capability.
> > > >
> > > > > [1] Zhang et al., "Discovering generalizable multi-agent coordination skills from multi-task offline data", ICLR 2022.

---

### Official Review · Reviewer_i2DM · 2023-06-23

**Soundness:** 3 good
**Presentation:** 3 good
**Contribution:** 3 good
**Rating:** 7
**Confidence:** 4

**Summary:**

This paper focuses on transfer learning of multi-agent reinforcement learning (MARL), by establishing generalizable sub-tasks to enable knowledge reuse. Empirical evidence underscores that the proposed algorithm demonstrates robust zero-shot generalization across a variety of tasks. Furthermore, the algorithm showcases ample transferability and surpasses contemporary methods in both multi-task and single-task problem areas.

**Strengths:**

(a) The design of the algorithm is presented convincingly and with remarkable clarity, providing an adequate level of detail.

(b) The paper presents compelling empirical results that display the algorithm's leading-edge performance in both multi-task and single-task scenarios. The case study outlined in Section 4.3 greatly aids in illuminating its superior capabilities.

(c) The exploration of transferable MARL represents a pivotal research area with substantial potential for practical applications.

**Weaknesses:**

(a) The paper seems to omit the related work section, which could be pivotal in comprehending the originality and contributions of this study.

(b) As it stands, the algorithm's generalization capacity is restricted to target tasks that can be broken down into a series of task-independent subtasks derived from source tasks. If the tasks are more diverse, it would be beneficial to consider task-specific subtasks as well.


**Questions:**

(a) The paper seems to address the same research problem as reference [36]. It would be beneficial to clarify your unique contributions and innovations in comparison.

(b) For zero-shot generalization to be applicable across multiple tasks, is it necessary for the tasks to be related or to follow the same distribution? If so, it's crucial to articulate this clearly.

(c) An overview of studies on multi-task skill-based (also known as hierarchical) MARL should be provided. More broadly, it would be useful to include work on multi-agent option (also known as skill) discovery in the related works section.

(d) The "Adaptive Subtask Semantics" section appears to heavily rely on the Action Semantics Network (ASN). Providing some background on ASN could aid in comprehension.

(e) In the multi-task settings, is the sole difference among tasks the number of entities, for example, shifting from 8m to 10m?

(f) Given its close relevance, it would be beneficial to include [36] as one of the baseline comparisons. (New results are not mandatory.)

**Limitations:**

The limitation part is not provided.

---

> ### Author Rebuttal · Authors · 2023-08-09
>
> We appreciate your positive review, insightful feedback, and constructive comments that help improve the quality of the paper! We are glad to answer your questions and would appreciate any further response.
>
> 1. > **Weakness (a)**: The paper seems to omit the related work section, which could be pivotal in comprehending the originality and contributions of this study.
>    >
>    > **Question (c)**: An overview of studies on multi-task skill-based (also known as hierarchical) MARL should be provided. More broadly, it would be useful to include work on multi-agent option (also known as skill) discovery in the related works section.
>    >
>    > **Question (d)**: The "Adaptive Subtask Semantics" section appears to heavily rely on the Action Semantics Network (ASN). Providing some background on ASN could aid in comprehension.
>
>    We added a section of related work to the appendix of the revised paper. For details, please see "**Weakness 1**" of our global response.
>
> 2. > **Weakness (b)**: As it stands, the algorithm's generalization capacity is restricted to target tasks that can be broken down into a series of task-independent subtasks derived from source tasks. If the tasks are more diverse, it would be beneficial to consider task-specific subtasks as well.
>
>    Enable the MARL model generalizable across more diverse tasks is a big problem. In the future, we will improve DT2GS by considering both task-independent and task-specific subtasks, to adapt to larger distribution shifts between source and target tasks.
>
>
>
> 3. > **Question (a)**: The paper seems to address the same research problem as reference ODIS[36]. It would be beneficial to clarify your unique contributions and innovations in comparison.
>    >
>    > **Question (f)**: Given its close relevance, it would be beneficial to include [36] as one of the baseline comparisons. (New results are not mandatory.)
>
>    DT2GS is quite different from ODIS. Specifically:
>
>    - DT2GS belongs to the online paradigm, while ODIS belongs to the offline paradigm. The different paradigms make the comparison between the two algorithms lack sufficient significance. To demonstrate this, we directly compare the experimental results from the ODIS paper [7] with the experimental results of DT2GS.
>
>      | Task     | DT2GS          | UPDeT       | UPDeT-l     | UPDeT-m     | ODIS       |
>      | -------- | -------------- | ----------- | ----------- | ----------- | ---------- |
>      | 3m       | **100 ± 1.3**  | 98.4 ± 4.0  | 71.0 ± 16.6 | 82.8 ± 16.0 | 98.4 ± 2.7 |
>      | 5m_vs_6m | **93.8 ± 7.0** | 50.0 ± 29.5 | 12.1 ± 12.6 | 17.2 ± 28.0 | 53.9 ± 5.1 |
>
>      As is shown in the table, the results of UPDeT-l, UPDeT-m, and ODIS come from the ODIS paper and are obtained from Expert offline data. We can observe that DT2GS outperforms ODIS in terms of asymptotic performance, especially on the hard task 5m_vs_6m. However, this performance difference cannot be used to prove which algorithm is better, as it is more likely caused by the paradigm to which the algorithms belong. Furthermore, it can be observed that the UPDeT in our paper (the column of UPDeT) is much better than the UPDeT in the ODIS paper (the columns of UPDeT-l and UPDeT-m), which further supports our conclusion.
>
>    - Different problem orientations. ODIS is designed to discover generalizable subtasks (skills) from multi-task offline data. Although DT2GS can interact with multiple tasks simultaneously, its original intention was to discover task-independent subtasks from a single source task and then generalize them to multiple target tasks.
>
>    - Different designs of subtask encoder. ODIS endows the subtask encoder with generalization by a data-driven approach, which prevents the subtask encoder from overfitting to some source task through multi-task offline data. Different from ODIS, DT2GS endows the subtask encoder with generalization by a structure-designed approach, which assigns subtask based on the agent's {subtask, entity-observation} history rather than {action, observation} history. But DT2GS does not conflict with multi-task, and even its generalization capability may be further improved due to simultaneous interaction with multi-tasks.
>
>    - DT2GS further studied the adaptive semantics of the same subtask on different tasks, which is important to further improve the model's generalization capability and was not mentioned in ODIS. The generalizable subtasks should maintain consistent yet scalable semantics across tasks. That is, the same subtask may have different manifestations on different tasks, which can be captured by constructing adaptive semantics of the subtask.
>
>
>
>
> 4. > **Question (b)**: For zero-shot generalization to be applicable across multiple tasks, is it necessary for the tasks to be related or to follow the same distribution? If so, it's crucial to articulate this clearly.
>
>    The zero-shot generalization scenario of DT2GS requires a moderate distribution shift between the source and target task. And we articulated this requirement in our revised paper.
>
>
>
> 5. > **Question (e)**: In the multi-task settings, is the sole difference among tasks the number of entities, for example, shifting from 8m to 10m?
>
>    In the 2 multi-task problems, including the *marine*-series tasks ({3m, 8m, 8m_vs_9m, 10m_vs_11m}) and *stalker_zealot*-series tasks ({2s3z, 3s5z, 3s5z_vs_3s6z}), the number of entities varies across tasks. However, we have demonstrated in Figure 1 that DT2GS contains sufficient transferability across tasks, in which the number and type of entities simultaneously shift among tasks.

---

> > ### Comment · Reviewer_i2DM · 2023-08-18
> >
> > I appreciate the effort made by the authors for this rebuttal. Most of my concerns are resolved. I would maintain my score at this stage.

---

### Official Review · Reviewer_5oAQ · 2023-07-05

**Soundness:** 3 good
**Presentation:** 1 poor
**Contribution:** 3 good
**Rating:** 5
**Confidence:** 4

**Summary:**

This work proposes DT2GS (Decompose a Task inTO a series of Generalizable Subtasks) that addresses multi-agent reinforcement learning in the contexts of zero-shot generalization, transfer, and multi-task. DT2GS learns task-independent subtasks that are characterized by the effects of each agent on itself and other agents when solving the subtask. The proposed method outperforms baselines in SMAC environment in zero-shot, transfer, and multi-task scenarios.

**Strengths:**

**S1. Intuitive and motivating formulation of subtasks**

The concept of defining a subtask based on the influence an agent on other entities is quite interesting and intuitive. This approach effectively translates into the context of multi-agent Reinforcement Learning (RL) problems, much like how the definition of a subtask is grounded on the Markov Decision Process (MDP) dynamics in single-agent RL problems. This parallel offers a refreshing perspective on tackling multi-agent RL scenarios. Also, it is well demonstrated in Figures 5 and 6.

**S2. Generalizing MARL**

The approach of expanding the scope of multi-agent Reinforcement Learning (RL) to incorporate zero-shot and multitask problems, achieved through the introduction of task-independent subtasks, is notably intriguing. Furthermore, the experiments have been designed to effectively demonstrate these problems.

**Weaknesses:**

**W1. Lack of clarity**

This paper could significantly benefit from substantial revisions to enhance its overall clarity.

- Concerning the technical writing, the manuscript suffers from the frequent use of undefined notations and interchangeable deployment of different notations, which create confusion. There are discrepancies between the notations used in equations and those in figures. For instance, the subscripts n, N, and n_a are inconsistently applied, leading to ambiguity. The varying entity notations, which also diverge from those used in ASN [1], generate unnecessary confusion, despite possibly facilitating the writing process. The clarity is further compromised when it is uncertain whether terms like o_i or o^t_i refer to the raw observation or the entity (see Line 73, Line 86, and Figure 1).
- The authors present the detailed definition of the subtasks after these terms have already been utilized in the text. To improve the flow and comprehension, it could be beneficial to relocate the definitions found on Line 143 to a position before Line 96, where they are first employed.
- The paper also contains lengthy sentences riddled with grammatical errors (e.g., Line 138 and Line 141), detracting from the readability.
- The description of the adaptive subtask semantics alongside the attention mechanism, specifically from Line 156 to Line 163, is notably vague and could use clearer explanations.

**W2. Lack of comparisons to previous work**

- The absence of a dedicated section for related work hinders the readers' ability to contextualize this study within the broader scope of the relevant literature. It is strongly suggested that the authors include a comprehensive summary of related work. This should highlight the strengths, weaknesses, differences, and improvements offered by the proposed method in comparison to other techniques in the field. Without this context, DT2GS may risk being perceived as merely a combination of ASN (entity) [1] and UPDeT (attention) [2].
- The authors appear to have overlooked the need to reference or empirically compare their work to other related studies that also focus on subtasks in Multi-Agent Reinforcement Learning (MARL). Notably missing comparisons to RoDE [3] and LDSA [4] could provide valuable benchmarks.

[1] Wang et al., “Action Semantics Network: Considering the Effects of Actions in Multiagent Systems”, ICLR 2020.

[2] Hu et al., “UPDeT: Universal Multi-agent Reinforcement Learning via Policy Decoupling with Transformers,” ICLR 2021.

[3] Wang et al., “RODE: Learning Roles to Decompose Multi-Agent Tasks,” ICLR 2021.

[4] Yang et al., “LDSA: Learning Dynamic Subtask Assignment in Cooperative Multi-Agent Reinforcement Learning,” NeurIPS 2022.


**Acknowledgment Following Rebuttal**

The author's rebuttal offered comprehensive revisions to the notations for enhanced clarity. Additionally, the inclusion of pseudocode aids in understanding the core concepts. I believe that implementing these proposed changes will significantly strengthen the work.

**Questions:**

**Q1.** The paper seems to lack a rigorous mathematical formulation that captures the definition of a subtask provided on Line 142. Could the authors clarify this aspect with a suitable formulation?

**Q2.** The two-stage self-attention mechanisms utilized in the subtask semantics module do not appear to be immediately intuitive. Could the authors further clarify their ideas from Line 146 to Line 163, particularly focusing on how this structure can prompt the subtask to categorize the impact of the agent on the entities?

**Q3.** It's somewhat perplexing why the MLP (Multi-Layer Perceptron) is designated as "similarity", considering it doesn't directly compute the similarities of the embeddings. Could the authors clarify this terminology?

**Q4.** It’s unclear how the learned subtask context is actually used by the policy. A pseudocode
of the entire process including MAPPO update would be helpful.

**Q5.** The depiction of multiple encoders with a single decoder in Figure 1 is rather confusing. Given that it's a decentralized system, it seems unlikely that each agent shares all input embeddings from all encoders. Does the figure represent the encoder and decoder for the n-th agent, where only the n-th embedding is input into the decoder, excluding the other n-1 agents?

**Q6.** Is the robustness of DT2GS dependent on the choice of n_k=4? Would the method's performance vary significantly with a different choice of n_k?

**Limitations:**

Despite the authors' affirmation in the checklist regarding limitations, neither the main manuscript nor the appendix seems to provide any explicit discussion of the potential limitations of the proposed method.

A possible limitation of the proposed work could be its inability to generalize to target tasks with a larger distribution shift, especially those that necessitate the implementation of entirely novel subtasks. For instance, an agent that learns four subtasks during training may struggle when the target task requires the execution of a fifth, previously unseen subtask. Moreover, the computational and memory demands imposed by the subtask encoder and semantics module could also present significant challenges, serving as additional potential limitations of this work

---

> ### Author Rebuttal · Authors · 2023-08-10
>
> Thanks a lot for your advice on further improving this paper. We would like to discuss them one by one. Any further discussion will be appreciated.
>
> 1. > **Weakness 1**: Lack of clarity
>
>    We revised our writing. Specifically :
>
>    (1) We standardized the use of subscripts in our revised paper as follows:
>
>    - $n$ -- the number of agents.
>
>    - $m$ -- the number of entities.
>
>    - $n_{ally}$ -- the number of allies
>
>    - $n_{enemy}$ -- the number of enemies
>
>    In general, the following equation holds: $n=n_{ally}+1, m=n+n_{enemy}$. And we also standardized the use of the following terms to make our paper clearer
>
>    - $o_i$ -- the raw observation of agent $i$
>
>    - $o_i^t$ -- the raw observation of agent $i$ at timestep $t$ (that is, the superscript $t$ denotes timestep)
>
>    - $o_{i,j}$ -- the entity-observation of agent $i$ on entity $j$
>
>    - $o_{i,j}^{t}$ -- the entity-observation of agent $i$ on entity $j$ at timestep $t$
>
>    (2) We added the definition of subtask semantics, which refers to the effects of an agent on entities when performing a given subtask, in the first paragraph of Section 3 (around Line 100).
>
>    (3) We revised the grammatical errors and broke down lengthy sentences into shorter ones that are easier to understand.
>
> 2. > **Weakness 2**: Lack of comparisons to previous work
>
>    (1) We added a section of related work to the appendix of the revised paper. For details, please see "**Weakness 1**" our global response. Besides, It should be emphasized that DT2GS is not a combination of ASN and UPDeT. The innovation of DT2GS includes proposing a scalable subtask encoder and an adaptive subtask semantic module, both of which are not proposed by previous work.
>
>    (2) We added RODE, ROMA, and HSD into our reference, and added RODE, ROMA, HSD, and LDSA as baselines in our experiments. The results can be seen in "**Enhancement 1**" of our global response. We see that DT2GS outperforms all baselines in terms of asymptotic performance.
>
>
> 3. > **Question 1**: A formula for subtask
>
>    The mathematical formulation of the subtask is: $k_i^t=\text{Gumbel-Softmax}(h_i^t)$
>
>    where $h_i^t$ coming from Formula (4) is the history representation of agent $i$. We added the mathematical formulation of the subtask around Line 132 of our paper to make it more clear.
>
> 4. >**Weakness 1 (4) && Question 2**: Clarify for subtask semantics
>
>    We made the following improvements to make the part of Adaptive Subtask Semantics more clear to read in our revised paper.
>
>    First, we set the one-hot subtask embedding $z_i^t$ as a formula: $z_i^t=\text{Embedding}(k_i^t)$
>
>    Then, we modified Formula (5) as follows: $\phi_i^t=\text{softmax}(\frac{\hat{Q}_i^tK_i^t}{\sqrt{d_K}})V_i^t, \quad \hat{Q}_i^t=W_Qz_i^t$
>
>    where $K_i^t = W_{K}[o_{i, 1}^t, o_{i, 2}^t, ..., o_{i, m}^t], \quad V_i^t = W_{V}[o_{i, 1}^t, o_{i, 2}^t, ..., o_{i, m}^t]$
>
>    Therefore, we model the adaptive subtask semantics as a weighted sum of all entity-observations to differentiate the impact of the agent on entities.
>
>    At the same time, we kept Formula (5) unchanged: $\varphi_i^t=\text{softmax}(\frac{{Q}_i^tK_i^t}{\sqrt{d_K}})V_i^t, \quad {Q}_i^t=W_Qo_i^t$
>
>    Therefore, the calculation of adaptive subtask semantics $\phi_i^t$ and action semantics $\varphi_i^t$ can be differentiated by its query matrix $\hat{Q}_i^t$ and $Q_i^t$.
>
>    Besides, we modified the $Q(a_j|o_i^t)$ in Formula (7) to $Q_{value}(a_j|o_i^t)$ to differentiate the action-value function ($Q_{value}$) and the query matrix ($Q_i^t$ or $\hat{Q}_i^t$) in the Attention mechanisms.
>
>    In addition, we also added  $Q_{value}$ and $Pr$ in Figure 3 to facilitate readers in understanding the formulas by referring to Figure 3.
>
>
>
> 5. > **Question 3**: Using MLP for similarity.
>
>    In fact, the uses of MLP are diverse. MLP is also a kind of embedding, which can describe similarity to a certain extent, for example, there is a similar usage in [1]. Our experimental results also prove the effectiveness of MLP as similarity to a certain extent.
>
>    > [1] Zeng, Kuo-Hao, et al. "Moving Forward by Moving Backward: Embedding Action Impact over Action Semantics", ICLR 2022.
>
>
>
> 6. > **Question 4**: How subtask is used by policy.
>
>    We believe that the combination of Figure 1 and Figure 3 serves as pseudo code. But to provide readers with a clearer understanding, we also included pseudo code in the revised paper's appendix.
>
>
>
> 7. > **Question 5**: Multiple encoders with single decoder in Figure 1.
>
>    Actually, our encoder and decoder can be shared or independent among all agents. Sorry for the inconvenience caused. And we modified Figure 1 in our revised paper to avoid causing such confusion to readers.
>
>
>
> 8. > **Question 6**: the choice of $n_k=4$
>
>    $n_k$ is the number of task-independent subtasks across a series of tasks. As is shown in the ablation experiment we conducted in the appendix, increasing $n_k$ appropriately can enhance the diversity of subtasks and thus capture all task-independent subtasks as much as possible. However, an excessive value of $n_k$ can also affect the efficiency of our method, resulting in a poorer performance. Therefore, it's natural that a reasonable choice of $n_k$ matters to DT2GS.
>
>
>
> 9. > **Limitations:** Larger distribution shift between source and target tasks.
>
>    DT2GS can capture all of the task-independent subtasks, which are generalizable across different tasks. Therefore, the generalization scenario of DT2GS is that the distribution shift between the source and target task is moderate. And we added this limitation in our revised paper. Furthermore, expanding the applicability of generalization scenarios and reducing computational complexity are the focal points of future work.

---

> > ### Comment · Reviewer_5oAQ · 2023-08-10
> > **Response to the Rebuttal**
> >
> > I appreciate the author's detailed responses. I believe that the manuscript's clarity will greatly benefit from the upcoming clarifications.
> >
> > With respect to Question 4, some elements of the algorithm's execution remain unclear. Specifically, Figures 1 and 3 are individually complex, making it challenging for me to translate their combination into pseudocode. I would be grateful if the authors could provide detailed pseudocode with revised notations, which would significantly aid comprehension of the algorithm.

---

> > > ### Author Response · Authors · 2023-08-11
> > > **Thank you for your reply!**
> > >
> > > The overall pseudocode of DT2GS based on MAPPO is as follows:
> > >
> > > **Algorithm 1: DT2GS based on MAPPO**
> > >
> > > ---
> > >
> > > ​            **Input:** The parameters $\theta_{En}^{\pi}$ for Scalable Subtask Encoder of actor $\pi$; the parameters $\theta_{De}^{\pi}$ for Adaptive Action Decoder of actor $\pi$; the parameters $\theta^{V}$ for critic $V$; the number $n_k$ for subtasks
> > >
> > > ​            **Output:** The Scalable Subtask Encoder $\pi_{En}$ and Adaptive Action Decoder $\pi_{De}$ for actor $\pi$; the critic $V$
> > >
> > > ​		&ensp;1: &emsp;Initialize $\theta_{En}^{\pi}, \theta_{De}^{\pi}, \theta^{V}, n_k$
> > >
> > > ​        	&ensp;2: &emsp;Initialize the total timesteps $step_{\text{max}}$ interaction with environment; the total timesteps $T$ of an episode; the number of episodes $batch\\_size$ for each actor/critic update
> > >
> > > ​        	&ensp;3: &emsp;Initialize $step \leftarrow 0$
> > >
> > > ​		&ensp;4: &emsp;**while** $step \leq step_{\text{max}}$
> > >
> > > ​		&ensp;5: 		   &emsp;&emsp;set data buffer $D$ = \{\}
> > >
> > > ​                &ensp;6:     	&emsp;&emsp;**for** $idx=1$ **to** $batch\\_size$ **do**
> > >
> > > ​                &ensp;7:     		&emsp;&emsp;&emsp;$\tau=[]$ empty list
> > >
> > > ​                &ensp;8:     		&emsp;&emsp;&emsp;initialize actor RNN states $h_{1, \pi}^{0}, ... , h_{n, \pi}^{0}$ for each agent
> > >
> > > ​                &ensp;9:        	 &emsp;&emsp;&emsp;initialize critic RNN states $h_{1, V}^{0}, ... , h_{n, V}^{0}$ for each agent
> > >
> > > ​             10:         	&emsp;&emsp;&emsp;initialize subtasks $k_1^0, ..., k_n^0$ for each agent   // ***Initialize subtasks with $n_k$-dim zero vector***
> > >
> > > ​              11:       	  &emsp;&emsp;&emsp;**for** timestep $t=1$ **to** $T$ **do**
> > >
> > > ​              12:   	   	  	 &emsp;&emsp;&emsp;&emsp;**for all** agents $i$ **do**
> > >
> > > ​              13:             			&emsp;&emsp;&emsp;&emsp;&emsp;$k_i^t, h_{i, \pi}^t = \pi_{En}(o_i^t, k_i^{t-1}, h_{i, \pi}^{t-1};\theta_{En}^{\pi})$  // ***Call for Scalable Subtask Encoder Module***
> > >
> > > ​          14:             			&emsp;&emsp;&emsp;&emsp;&emsp;$a_i^t = \pi_{De}(o_i^t, k_i^t; \theta_{De}^{\pi})$   // ***Call for Adaptive Action Decoder Module***
> > >
> > > ​          15:             			&emsp;&emsp;&emsp;&emsp;&emsp;$v_i^t, h_{i, V}^t = V(s_i^t, h_{i, V}^{t-1}; \theta^V)$
> > >
> > > ​          16:         			&emsp;&emsp;&emsp;&emsp;**end for**
> > >
> > > ​          17:         			&emsp;&emsp;&emsp;&emsp;Execute actions $\pmb{a^t}$, observe $r^t, s^{t+1}, \pmb{o^{t+1}}$
> > >
> > > ​          18:         			&emsp;&emsp;&emsp;&emsp;$\tau += [s^t, \pmb{o^t}, \pmb{h_{\pi}^t}, \pmb{h_V^t}, \pmb{k^t}, \pmb{a^t}, r^t, s^{t+1}, \pmb{o^{t+1}}]$
> > >
> > > ​          19:   	   	&emsp;&emsp;&emsp;**end for**
> > >
> > > ​          20:          	&emsp;&emsp;&emsp;$step += T$
> > >
> > > ​          21:    	  	&emsp;&emsp;&emsp;Compute advantage estimate $\hat{A}$ via GAE on $\tau$
> > >
> > > ​          22:    	  	&emsp;&emsp;&emsp;Compute reward-to-go $\hat{R}$ on $\tau$
> > >
> > > ​          23:    	 	 &emsp;&emsp;&emsp;Split trajectory $\tau$ into chunks of length $L$
> > >
> > > ​          24:    	  	&emsp;&emsp;&emsp;**for** $l=0,L,2L,...,(T//L)*L$ **do**
> > >
> > > ​          25:         		  &emsp;&emsp;&emsp;&emsp;$D = D \cup (\tau[l:l+L], \hat{A}[l:l+L], \hat{R}[l:l+L])$
> > >
> > > ​          26:        	  &emsp;&emsp;&emsp;**end for**
> > >
> > > ​	      27:          &emsp;&emsp;**end for**
> > >
> > > ​          28: 	     &emsp;&emsp;**for** mini-batch $k=1, ... ,K$ **do**
> > >
> > > ​          29: 	  		&emsp;&emsp;&emsp;$b \leftarrow$ random mini-batch from $D$ with all agent data
> > >
> > > ​          30:          	&emsp;&emsp;&emsp;**for** each data chunk $c$ in the mini-batch $b$
> > >
> > > ​          31:       			&emsp;&emsp;&emsp;update RNN hidden states for $\pi_{En}$ and $V$ from first hidden state in data chunk
> > >
> > > ​          32:     	 	&emsp;&emsp;&emsp;**end for**
> > >
> > > ​          33: 	 	&emsp;&emsp;**end for**
> > >
> > > ​          34:  	    &emsp;&emsp;Adam update $\theta_{En}^{\pi}, \theta_{De}^{\pi}$ on actor loss with data $b$
> > >
> > > ​          34:  	    &emsp;&emsp;Adam update $\theta^V$ on critic loss with data $b$
> > >
> > > ​		  36: &emsp;**end while**
> > >
> > > ​          37: &emsp;**Return** $\pi_{En}, \pi_{De}, V$

---

> > > > ### Author Response · Authors · 2023-08-11
> > > > **Thank you for your reply!**
> > > >
> > > > And the pseudocode of Scalable Subtask Encoder Module and Adaptive Action Decoder Module used in DT2GS are as follows:
> > > >
> > > > **Algorithm 2: Scalable Subtask Encoder**
> > > >
> > > > -----
> > > >
> > > > ​            **Input:** The parameters $\theta_{En}^{\pi}$ for Scalable Subtask Encoder of actor $\pi$, agent $i$'s observation $o_i^t$ in timestep $t$, agent $i$'s subtask $k_i^{t-1}$ and actor RNN state $h_{i, \pi}^{t-1}$ in timestep $t-1$
> > > >
> > > > ​			**Output:** agent $i$'s subtask $k_i^t$ and actor RNN state $h_{i, \pi}^t$ in timestep $t$
> > > >
> > > > ​            1: &emsp;Compute entity-embedding $(\mu_{i,j}^t, \sigma_{i,j}^t), j=1,...,m$ by Formula (2)
> > > >
> > > > ​            2: &emsp;Compute observation-embedding $e_i^t$ by Formula (3)
> > > >
> > > > ​            3: &emsp;Compute actor RNN state $h_{i, \pi}^t$ by Formula (4)
> > > >
> > > > ​            4: &emsp;Compute subtask $k_i^t$ with $k_i^t \sim \text{Gumbel-Softmax}(h_{i, \pi}^t)$
> > > >
> > > > ​            5: &emsp;**Return** $k_i^t, h_{i, \pi}^t$
> > > >
> > > >
> > > >
> > > >
> > > >
> > > > **Algorithm 3: Adaptive Action Decoder**
> > > >
> > > > -----
> > > >
> > > > ​            **Input:** The parameters $\theta_{De}^{\pi}$ for Adaptive Action Decoder of actor $\pi$, agent $i$'s observation $o_i^t$ and subtask $k_i^{t}$ in timestep $t$
> > > >
> > > > ​            **Output:** agent $i$'s action $a_i^t$ in timestep $t$
> > > >
> > > > ​            1: &emsp;Compute subtask embedding $z_i^t$ with $z_i^t=\text{Embedding}(k_i^t)$
> > > >
> > > > ​            2: &emsp;Compute adaptive subtask semantics $\phi_i^t$ by Formula (5)
> > > >
> > > > ​            3: &emsp;Compute action semantics $\varphi_i^t$ by Formula (6)
> > > >
> > > > ​            4: &emsp;Sample action $a_i^t$ from the actions' probability distribution $Pr_i^t$ computed by Formula (7).
> > > >
> > > > ​            5: &emsp;**Return** $a_i^t$
> > > >
> > > > As shown in the pseudocode, we initialize subtasks with $n_k$-dim zero vector at the start of each episode. And at each timestep, the Scalable Subtask Encoder and the Adaptive Action Decoder are utilized in the actor to obtain each agent's subtask and action, respectively.
> > > >
> > > > Hope that these pseudocode can help you understand DT2GS more clearly.

---

> > > ### Author Response · Authors · 2023-08-18
> > > **Sincerely invite for further discussion.**
> > >
> > > Dear reviewer, as the discussion stage draws to a close, we would greatly appreciate your opinions. Are there any additional concerns or suggestions that you would like to share? We are more than willing to engage in a constructive discussion with you.

---

> > > > ### Comment · Reviewer_5oAQ · 2023-08-19
> > > >
> > > > I appreciate the authors for providing detailed pseudocode, which significantly aids in understanding the paper.  I trust that the authors will improve the clarity of the paper, as indicated in their response. Based on this, I am raising my score to a 'borderline accept'. However, I am not inclined to raise my score to a 'clear accept' at this stage. This is due to the extensive rewriting that the paper requires to address the proposed changes, and the fact that I cannot review the revised manuscript during this period.

---

> > > > > ### Author Response · Authors · 2023-08-19
> > > > >
> > > > > Thanks for your valuable suggestions that have made our paper clearer! And our revised manuscript incorporated all the improvements mentioned during the rebuttal period, including clearer symbol definitions, clearer definitions and calculations of subtasks and their semantics, more comprehensive experimental results with more baselines, well-organized related work, and a clearer presentation of Figure 1 using a single encoder and single decoder.

---

### Official Review · Reviewer_McXb · 2023-07-07

**Soundness:** 4 excellent
**Presentation:** 3 good
**Contribution:** 3 good
**Rating:** 7
**Confidence:** 3

**Summary:**

The paper introduces the DT2GS framework to improve the generality of agents in Multi-Agent Reinforcement Learning (MARL) by decomposing a task into generalizable subtasks. The authors use a scalable subtask encoder to identify appropriate subtasks based on historical entity-observation pairs, instead of action-observation pairs, which enhances generalizability. With the subtask selected by the scalable subtask encoder, the paper employs an adaptive subtask semantics with a self-attention mechanism to compute adaptive action semantics that are then passed to an MLP to obtain action distribution and Q values. This adaptive action semantics contains information about the current subtask selected as well as observations from all the entities, making it sufficient for computing action and Q values. The self-attention mechanism used does not have a position embedding, which is suited to the requirement of scalability across tasks regardless of the number of entities and permutation invariance. Empirical results demonstrate that DT2GS possesses sound zero-shot generalization capability across tasks, exhibits sufficient transferability, and outperforms existing methods in both multi-task and single-task problems.

**Strengths:**

1. The proposed framework's structure is well suited to the problem setting, including the scalable subtask encoder and the adaptive subtask semantics.
2. Experimental results are promising, demonstrating improved performance compared to baseline methods, as well as providing insightful analysis on subtask percentage.

**Weaknesses:**

1. A more detailed introduction of related work could be helpful for readers if space allows, such as the network design or methodology of ASN.
2. In Figure 1, the right part uses "Env Cognition" and "Cognition Encoder: MLP". Adding the word "cognition" to appropriate locations in the text around Equation 2 would make Figure 1 match the text better.

**Questions:**

Figure 4 only reports the performance of each algorithm trained on the source task but tested on the target task. What is the performance of algorithms on the source task? If one algorithm performs poorly even on the source task, then it makes little sense to test it on a different target task.

**Limitations:**

The authors did not explicitly discuss the limitations of their work in the paper. However, no major limitations or potential negative societal impacts have been identified in this review.

---

> ### Author Rebuttal · Authors · 2023-08-08
>
> Thanks for your detailed review. We are glad to discuss your concerns one by one. Any further discussion will be appreciated.
>
> 1. > **Weakness 1**: A more detailed introduction of related work could be helpful for readers if space allows, such as the network design or methodology of ASN.
>
>    We added a section of related work to the appendix of the revised paper. For details, please see **Weakness 1** of our global response.
>
> 2. > **Weakness 2**: In Figure 1, the right part uses "Env Cognition" and "Cognition Encoder: MLP". Adding the word "cognition" to appropriate locations in the text around Equation 2 would make Figure 1 match the text better.
>
>    We added "cognition" in our revised paper to ensure consistency between text and figure. Specifically, we modified the content of Line 125 to "and the observation embedding $e_i^t$, which is also referred to as Env Cognition of agent $i$ in Figure 1, is constructed as: ".
>
> 3. > **Question**: Figure 4 only reports the performance of each algorithm trained on the source task but tested on the target task. What is the performance of algorithms on the source task? If one algorithm performs poorly even on the source task, then it makes little sense to test it on a different target task.
>
>    The asymptotic performance of DT2GS, UPDeT, and ASN_G are shown in the table as follows (values before '/' are the algorithms' performance of test winning rate on the source task, and values after '/' are the zero-shot performance of the model on target tasks). As we can see, although DT2GS has the same/similar asymptotic performance as UPDeT/ASN_G on all/most source tasks, its zero-shot generalization performance is significantly higher than UPDeT/ASN_G, achieving an average test winning rate surpass of about 22%/34% over all 8 zero-shot generalization scenarios.
>
>    | source / target     | DT2GS     | UPDeT     | ASN_G         |
>    | ------------------- | --------- | --------- | ------------- |
>    | 3s_vs_4z / 3s_vs_5z | 1 / 0.391 | 1 / 0.203 | 0.109 / 0     |
>    | 2s3z / 3s5z         | 1 / 0.336 | 1/ 0.062  | 0.547 / 0.208 |
>    | 3s5z / 3s5z_vs_3s6z | 1 / 0.875 | 1 / 0.648 | 0.969 / 0.547 |
>    | 8m / 8m_vs_9m       | 1 / 0.734 | 1 / 0.391 | 0.938 / 0.141 |
>    | 8m / 10m_vs_11m     | 1 / 0.609 | 1 / 0.539 | 0.938 / 0.188 |
>    | 8m / 25m            | 1 / 0.367 | 1 / 0.172 | 0.938 / 0.039 |
>    | 8m_vs_9m / 25m      | 1 / 0.484 | 1 / 0.234 | 0.469 / 0.164 |
>    | 8m_vs_9m / 5m_vs_6m | 1 / 0.180 | 1 / 0.070 | 0.469 / 0     |

---

> > ### Comment · Reviewer_McXb · 2023-08-16
> > **Thanks for your reply.**
> >
> > Thank you for submitting your rebuttal. The authors have made a commendable effort to address and answer my concerns, which is greatly appreciated.

---

### Author Rebuttal · Authors · 2023-08-10

1. > **Weakness 1**: Lack of a section for related work.

   Due to space limitations, we only provided a brief introduction to the related work in the Introduction section. Based on the feedback received, we added a detailed section of related work to the appendix of the revised paper. And the related work is organized according to the following structure: Firstly, we classify the methods of transfer learning across tasks in online MRAL into two categories: Network-design-based methods [1, 2, 3] and Task-embedding-based methods [4, 5, 6]. Secondly, we also summarized the methods of transfer learning across tasks belonging to offline MARL [7] and Curriculum Learning [8, 9] in MRAL. Thirdly, we summarized the related work in MARL focusing on concepts like skills/options/roles [10, 11, 12, 13], which are similar to subtasks studied in our method. Finally, we explained in detail the network design of ASN [2], the action semantics mentioned in ASN, and the population-invariant network (PIN) structure developed in UPDeT, which can help readers to understand the pipeline of our method.

   Previous works focusing on multi-agent transfer learning based on network-design or task-embedding, but lack of efficient knowledge reuse leading to the limited transferability. Our work leverages the knowledge reuse by ensuring semantic consistency between diverse tasks and prevents over-fitting of source tasks, which greatly improves transferability and generalization capability. Results show that DT2GS model can achieve zero-shot generalization capability and more robust transferability, achieving an average transfer speedup of 100×.

   > [1] Hu et al., “UPDeT: Universal Multi-agent Reinforcement Learning via Policy Decoupling with Transformers,” ICLR 2021.
   >
   > [2]  Wang et al., “Action Semantics Network: Considering the Effects of Actions in Multiagent Systems”, ICLR 2020.
   >
   > [3] Agarwal et al., "Learning transferable cooperative behavior in multi-agent teams", ArXiv 2019.
   >
   > [4] Liu et al., "Value function transfer for deep multi-agent reinforcement learning based on n-step returns", IJCAI 2019.
   >
   > [5] Qin et al., "Multi-agent policy transfer via task relationship modeling", ArXiv 2022.
   >
   > [6] Schäfer et al., "Learning task embeddings for teamwork adaptation in multi-agent reinforcement learning", ArXiv 2022.
   >
   > [7] Zhang et al., "Discovering generalizable multi-agent coordination skills from multi-task offline data", ICLR 2022.
   >
   > [8] Long et al., "Evolutionary population curriculum for scaling multi-agent reinforcement learning", ICLR 2019.
   >
   > [9] Wang et al., "From few to more: Large-scale dynamic multiagent curriculum learning", AAAI 2020.
   >
   > [10] Yang et al., “LDSA: Learning Dynamic Subtask Assignment in Cooperative Multi-Agent Reinforcement Learning,” NeurIPS 2022.
   >
   > [11] Wang et al., “RODE: Learning Roles to Decompose Multi-Agent Tasks,” ICLR 2021.
   >
   > [12] Wang et al., "Roma: Multi-agent reinforcement learning with emergent", ICML 2020.
   >
   > [13] Yang et al., "Hierarchical cooperative multi-agent reinforcement learning with skill discovery", AAMAS 2020.

2. > **Enhancement 1**: Added experimental comparisons with some related work.

   We added RODE, ROMA, and HSD into our reference, and added RODE, ROMA, HSD, and LDSA as baselines in our experiments. The new results are shown in the table below.

   | task         | DT2GS       | UPDeT       | MAPPO       | ASN         | LDSA        | RODE        | ROMA        | HSD         |
   | ------------ | ----------- | ----------- | ----------- | ----------- | ----------- | ----------- | ----------- | ----------- |
   | 3s5z_vs_3s6z | 1.00 (0.02) | 1.00 (0.17) | 0.00 (0.23) | 0.84 (0.11) | 0.00 (0.00) | 0.00 (0.00) | 0.01 (0.03) | 0.00 (0.00) |
   | 5m_vs_6m     | 0.94 (0.07) | 0.50 (0.30) | 0.84 (0.06) | 0.06 (0.04) | 0.68 (0.11) | 0.48 (0.22) | 0.67 (0.12) | 0.00 (0.00) |
   | 6h_vs_8z     | 1.00 (0.03) | 0.91 (0.08) | 0.47 (0.18) | 0.00 (0.01) | 0.01 (0.02) | 0.03 (0.03) | 0.15 (0.17) | 0.00 (0.00) |
   | corridor     | 0.95 (0.05) | 0.91 (0.07) | 1.00 (0.02) | 0.09 (0.05) | 0.80 (0.13) | 0.00 (0.00) | 0.01 (0.04) | 0.02 (0.05) |

   The values in the table represent the mean and variance of the test winning rate. As we can see, DT2GS outperforms all baselines. Besides, a more intuitive graph regarding this result can be found in the PDF file we submitted.

3. > **Enhancement 2**: Added experiments on other environments.

   We conducted zero-shot experiments on the physical deception task (Spread) in the multi-agent particle world environments (MPE) [1]. The experimental results are as follows.

   | source/target | DT2GS           | UPDeT           | ASN_G           |
   | ------------- | --------------- | --------------- | --------------- |
   | 3/2           | -84.44 (5.85)   | -83.61 (2.31)   | -97.43 (8.67)   |
   | 3/3           | -170.88 (23.18) | -206.39 (9.53)  | -212.66 (7.11)  |
   | 3/4           | -265.79 (52.21) | -293.92 (14.58) | -309.32 (22.61) |
   | 3/5           | -371.11 (63.95) | -370.28 (31.05) | -459.51 (19.16) |
   | 3/6           | -472.74 (99.25) | -496.00 (72.95) | -507.66 (21.75) |

   For example, "3/4" in the column of "source/target" indicates that the source task is set by 3 agents and 3 landmarks, while the target task is set by 4 agents and 4 landmarks. And the values in the column of the algorithm represent the mean and variance of the episode rewards. As we can see, DT2GS outperforms UPDeT and ASN\_G in terms of zero-shot generalization capability, achieving an average episode reward surpass of about 17.05 and 44.32, respectively. And a more intuitive bar chart regarding this experiment can be found in the PDF file we submitted.

   > [1] Lowe, Ryan, et al., "Multi-agent actor-critic for mixed cooperative-competitive environments", NeurIPS 2017.

---

### Decision · Program_Chairs · 2023-09-21

**Decision:**

Accept (poster)

**Comment:**

This paper proposes a new architecture for zero-shot generalisation in multi-agent RL settings. The proposed architectures have two modules: 1) a subtask encoder that chooses what subtask to perform and 2) an action decoder that selects an action conditioned on the subtask encoder's output. The result on StarCraft Multi-Agent Challenge shows that the proposed method not only generalises better to unseen tasks than relevant baselines but also achieve state-of-the-art performances on single-task and multi-task settings.

The reviewers agreed that the proposed architecture is interesting and reasonable, and the empirical results are quite strong. The reviewers' concerns around the clarity and the lack of discussion on the related work were mostly addressed by the rebuttal. In addition, the authors added more results during the rebuttal period, which further strengthened the result. Since all of the reviewers are not strongly against the paper, I recommend accepting this paper.